# Dual-mechanism based CTLs infiltration enhancement initiated by Nano-sapper potentiates immunotherapy against immune-excluded tumors

Yukun Huang[1], Yu Chen[1], Songlei Zhou[1], Liang Chen[1], Jiahao Wang[1], Yuanyuan Pei[1], Minjun Xu[1], Jingxian Feng[1], Tianze Jiang[1], Kaifan Liang[1], Shanshan Liu[1], Qingxiang Song[2], Gan Jiang [2], Xiao Gu[2], Qian Zhang[2], Xiaoling Gao [2]* & Jun Chen[1,3]*

The failure of immunotherapies in immune-excluded tumor (IET) is largely ascribed to the void of intratumoral cytotoxic T cells (CTLs). The major obstacles are the excessive stroma, defective vasculatures and the deficiency of signals recruiting CTLs. Here we report a dual-mechanism based CTLs infiltration enhancer, Nano-sapper, which can simultaneously reduce the physical obstacles in tumor microenvironment and recruiting CTLs to potentiate immunotherapy in IET. Nano-sapper consists a core that co-loaded with antifibrotic phosphates-modified α-mangostin and plasmid encoding immune-enhanced cytokine LIGHT. Through reversing the abnormal activated fibroblasts, decreasing collagen deposition, normalizing the intratumoral vasculatures, and in situ stimulating the lymphocyte-recruiting chemoat-tractants expression, Nano-sapper paves the road for the CTLs infiltration, induces the intratumoral tertiary lymphoid structures, thus reshapes tumor microenvironment and potentiates checkpoint inhibitor against IET. This study demonstrates that the combination of antifibrotic agent and immune-enhanced cytokine might represent a modality in promoting immunotherapy against IET.

[1] Key Laboratory of Smart Drug Delivery, Ministry of Education, School of Pharmacy, Fudan University, Lane 826, Zhangheng Road, Shanghai 201203, P.R. China. [2] Department of Pharmacology, Institute of Medical Sciences, Shanghai Jiao Tong University School of Medicine, 280 South Chongqing Road, Shanghai 200025, P.R. China. [3] Department of Pharmacy, Shanghai Pudong Hospital, Fudan University, 2800 Gongwei Road, Shanghai 201399, P.R. China. *email: shellygao1@sjtu.edu.cn; chenjun@fudan.edu.cn

Tumors that elude infiltration by cytotoxic T lymphocytes (CTLs) are termed immune-excluded tumor (IET)[1], which are particularly resistant to multiple types of treatment. IET covers the most detrimental malignancies including pancreatic ductal adenocarcinoma (PDAC), breast cancer[2], and ovarian cancer[3]. The 5-year overall survival rate of IET, take PDAC as an example, is around 5%[4,5]. Emerging advances in immunotherapy including immune-checkpoint blockade (ICB) have shown promise in melanoma, lung cancer and urothelial carcinoma[6]. However, such successes have not yet been translated to IET[7,8]. Both clinical and research evidences revealed that tumors with infiltrated CTLs could largely response to ICB[9], while IET, which is lack of CTLs, failed to response to ICB. Therefore, it is essential to enhance the infiltration of CTLs for successful ICB therapies in IET.

The immunosuppressive tumor microenvironment (TME) of IET preferentially restricts the infiltration of CTLs and several mechanisms could be hold responsible for. First, the cancer-associated fibroblasts (CAFs) suppress antitumor immunity by restricting T cells to stroma and preventing them from accumulating in the vicinity of cancer cells through the production of dense extracellular matrix (ECM) such as collagen and hyaluronan, as well as through the secretion of suppressive cytokines such as TGF-β and CXCL12[10]. Second, the defect tumor vasculatures delivers inadequate blood and reduce the transmigration of lymphocytes. Moreover, the anergic tumor endothelium lacks adhesion molecules such as ICAM and VCAM[11], and therefore represents one of the substantial rate-limiting barriers restraining anti-cancer immunity[12–14]. Third, the scarcely expressed recruiting signals of lymphocytes in IET, such as CCL19, CCL21, CXCL9, and CXCL10, have delayed the infiltration of CTLs[15–19]. Given the biophysical features of IET, obstacles that stop CTLs from infiltrating could be classified as "physical obstacles" which includes CAFs, ECM and defect tumor vasculatures, and "signal deficiency" which means the lack of recruiting cytokines.

In view of these features, two typical strategies for promoting the infiltration of CTLs have been developed, the direct modulation and indirect stimulation. For direct modulation, stroma attenuation (PEGPH20, pegylated recombinant human hyaluronidase, NCT03634332; AMD3100, Plerixafor, NCT03277209) and vessel normalization (bevacizumab, anti-VEGF monoclonal antibody, NCT02715531) are currently in clinical trials. PEGPH20 degrades hyaluronan in the tumor, AMD3100 inhibits the activity of CXCR4+ CAFs and bevacizumab improves the immune functionalities of surviving vessels. These therapies directly remodel the TME and allow the infiltration of CTLs. As for indirect stimulation, GVAX vaccine (NCT02243371) promotes the intratumoral recruitment of CTLs by inducing peripheral T cell activation. However, clinical outcomes so far are not satisfying, and there are several possible causes for these. On one hand, these therapies are not able to simultaneously cope with all obstacles such as CAFs, ECM and defect tumor vasculatures, which allow the compensation effects within the TME delay the CTLs infiltration. On the other hand, the insufficient recruiting signals in tumor still restrain the infiltration and it has been proved that once the chemoattractant for T cells is introduced, the antitumor potential of adoptive T cells is greatly enhanced[20]. In addition, the infiltrated immune suppressive cells accompanied with the physical obstacles reduction should also be addressed[21]. Taken together, these investigations demonstrated that neither direct modulation nor indirect stimulation is satisfying, and highlights the necessity of strategy design and rational drug selection.

To better enhance the infiltration of CTLs in IET, here we propose an infiltration enhancement strategy based on two mechanisms which synergistically breaks the physical obstacles and increases the recruiting signals. This strategy includes a natural small molecule that can reverse the abnormal activated CAFs and decrease collagen deposition, and an immune-enhanced cytokine can normalize defect vessels and stimulating recruiting signals. Specifically, PDAC was chosen as the representative IET model. α-mangostin (α-M), a natural xanthone isolated from the pericarps of mangosteen was selected since it had been reported to reduce liver fibrogenesis without obvious hepatotoxicity[22]. Therefore, we hypothesized it might be also effective in PDAC. However, the application of α-M was hindered by its poor water solubility and rapid elimination. Thus, we synthesized the phosphate of α-M (MP) to facilitate its delivery as phosphorylation can increase the bioavailability of hydrophobic drugs and phosphate esters could be cleaved by phosphatases in tumors[23–25]. LIGHT (tumor necrosis factor superfamily 14, TNFSF14, CD258) is an pleiotropic inflammatory cytokine which can promotes vascular inflammation, and facilitates T cell recruitment and activation[26]. LIGHT has been reported to normalize the intratumoral vessels, promote antitumor immunity through stimulating the secretion of chemoattractants recruiting lymphocytes (CCL21 and CXCL13), inhibit Treg-mediated immune-supression[27–29], and induce the tertiary lymphoid structures (TLSs) to support the local generation of tumor-specific immune responses[29,30]. However, as a tumor necrosis factor, LIGHT are largely restrained by its systemic toxicity after parenteral administration[31]. Thus, we constructed the plasmid encoding LIGHT (pLIGHT) to realize carrier-assisted tumor-specific expression and stimulation. To realize the simultaneously delivery of MP and pLIGHT, we introduced the calcium phosphate liposome (CaP) techniques[32,33], which has been reported for efficient in vivo transfection and delivery of small molecules possesses phosphate groups[34,35]. Besides, an ECM glycoprotein (tenascin C) targeting peptide (FHK, FHKHKSPALSPV) was decorated to the exterior of CaP[36]. Tenascin-C is exclusively expressed in the stroma surround the neoplastic glands of PDAC, which is an ideal target for in vivo PDAC-homing[37,38]. Accordingly, we integrated MP and pLIGHT into the FHK peptide-decorated CaP (FHK-pLIGHT@CaMP) to realize the synergistical improvement of CTLs infiltration and enhance the response of IET to ICB.

We evaluated FHK-pLIGHT@CaMP using two orthotopic murine PDAC models. The FHK peptide decoration enhances the tumor retention of the loaded components. MP reverses the activation of CAFs, decreases collagen deposition and relieves the compressed vessels. The secreted LIGHT recovers the spatial distribution and functions of vessels, and stimulates the expression of lymphocyte-recruiting chemoattractants; MP and LIGHT synergistically improve the CTLs infiltration and induce TLSs in deeper region of PDAC. The further combination of FHK-pLIGHT@CaMP with α-PD-1 exhibited an enhanced tumor suppression than α-PD-1 single therapy. Considering the similarities, we named FHK-pLIGHT@CaMP as "Nano-sapper", because sappers have responsibilities in breaking through defensive barriers, repairing roads and bridges, building military bases, and finally paving the way for subsequent troops during the war. As described in Fig. 1, the proposed dual-mechanism based infiltration enhancement strategy would provide a promising approach in sensitizing IET to ICB therapy.

## Results

**The synthesis and characterization of MP**. To construct Nano-sapper, we first synthesized the MP by phosphorylation of hydroxyl groups of α-M with diethyl chlorophosphates (Supplementary Fig. 1a). Phosphorylation is a common technique in prodrug development and the phosphorylation of α-M would

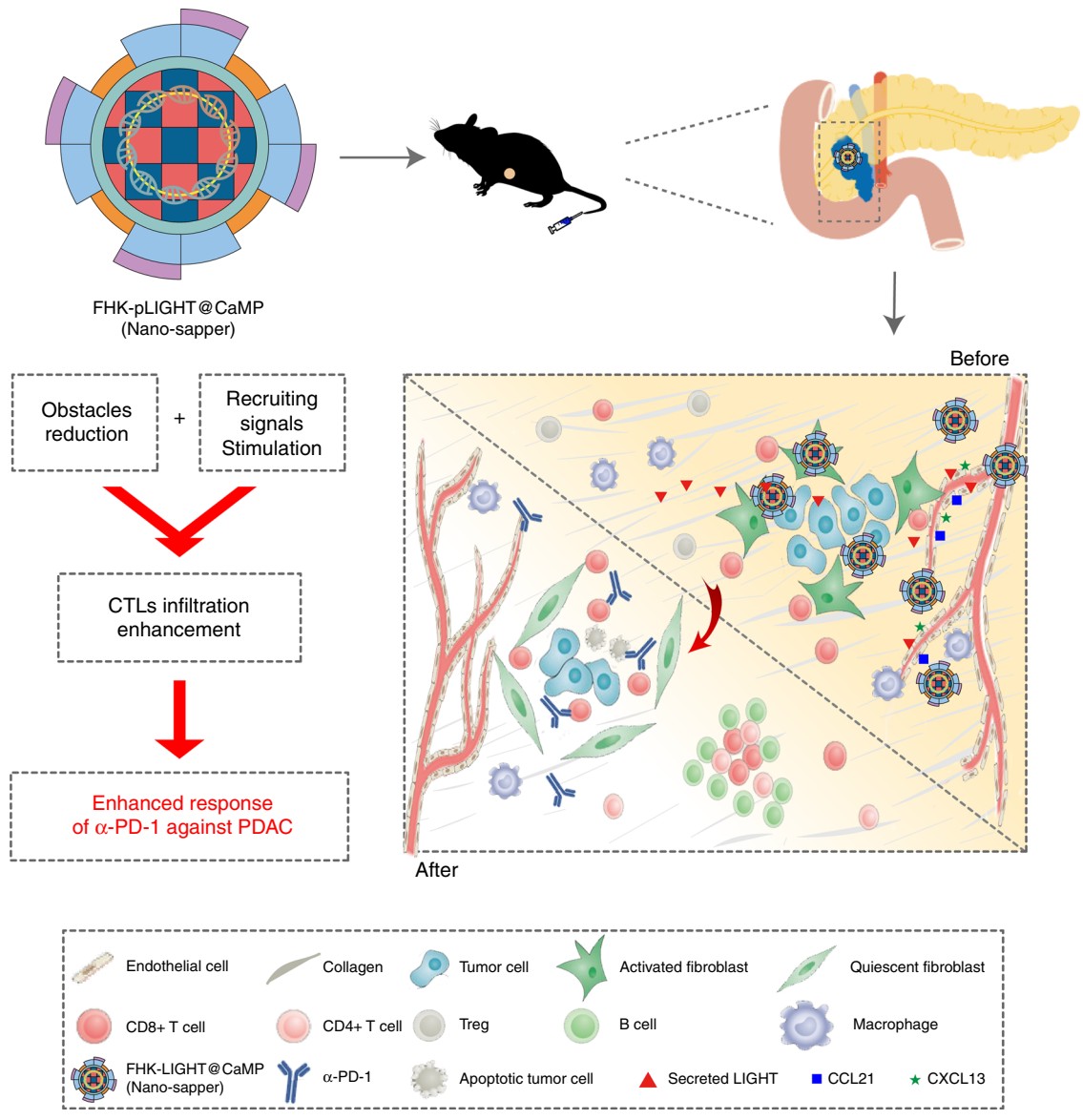

**Fig. 1 The expected effects of Nano-sapper synergized with immune-checkpoint inhibitor.** Nano-sapper specifically modulate the TME of PDAC, which involves the reduced physical barrier (attenuated stroma and normalized vessels) and the release of chemoattractants recruiting lymphocytes (CCL21 and CXCL13). Once the TME have been reprogrammed by Nano-sapper, a variety of adaptive immune cells migrate to the PDAC and enhance the anti-tumor effects of α-PD-1.

facilitate its precipitation with calcium to form amorphous core with efficient loading. The successful synthesis of MP was confirmed by mass spectrum ($m/z$ 571.0) $[M + H]^+$, $^1$H-NMR, $^{13}$C-NMR, and $^{31}$P-NMR (Supplementary Figs. 1b–9). We next validated the successful conversion of MP to α-M using alkaline phosphatase (AP) by HPLC. The retention time of the α-M in MP + AP solution was identical to the α-M standard, which possessed a dramatically increase comparing with the MP solution without AP (Supplementary Fig. 10a and b). We found that MP showed minimal cytotoxicity ($IC_{50} = 58.9$ μM) to activated NIH3T3 (Supplementary Fig. 10c) and could reverse the expression of CAFs related proteins including alpha-smooth muscle actin (α-SMA), fibroblast activation protein (FAP) and fibronectin to the basal level at the concentration of 27.8 μM (Supplementary Fig. 10d and e). We further explored the underlying impacts of MP on TGF-β/Smad signaling pathway which has been wildly recognized as the key factor in tumor fibrosis[39,40]. In detail, the activated NIH3T3 cells were incubated with varied concentration of MP for 24 h and the expression of

phospho-Smad2/3 (pSmad2/3) were determined. pSmad2/3 were gradually decreased accompanied with the increase of MP, and MP at 27.8 μM could reverse the enhanced pSmad2/3 expression in activated NIH3T3 cells (Supplementary Fig. 10f and g). These results revealed MP as a potent and safe reagent that could remodel CAFs.

**The preparation and characterization of Nano-sapper.** Nano-sapper was prepared as depicted in Fig. 2a. Briefly, the plasmid-loaded CaMP cores were synthesized via reversed-phase micro-emulsion, and then formed thin film with cholesterol, DOTAP, DSPE-PEG2000 and DSPE-PEG2000-FHK under reduced pressure. After that, the lipid film was hydrated with 10% sucrose solution to acquire Nano-sapper. The inner cores were precipitated through the interaction between calcium ions and MP/plasmids, and subsequently covered by asymmetric lipid bilayer with FHK peptide at the exterior. FHK-CaMP (without pLIGHT) and FHK-pLIGHT@CaP (without MP) were prepared through

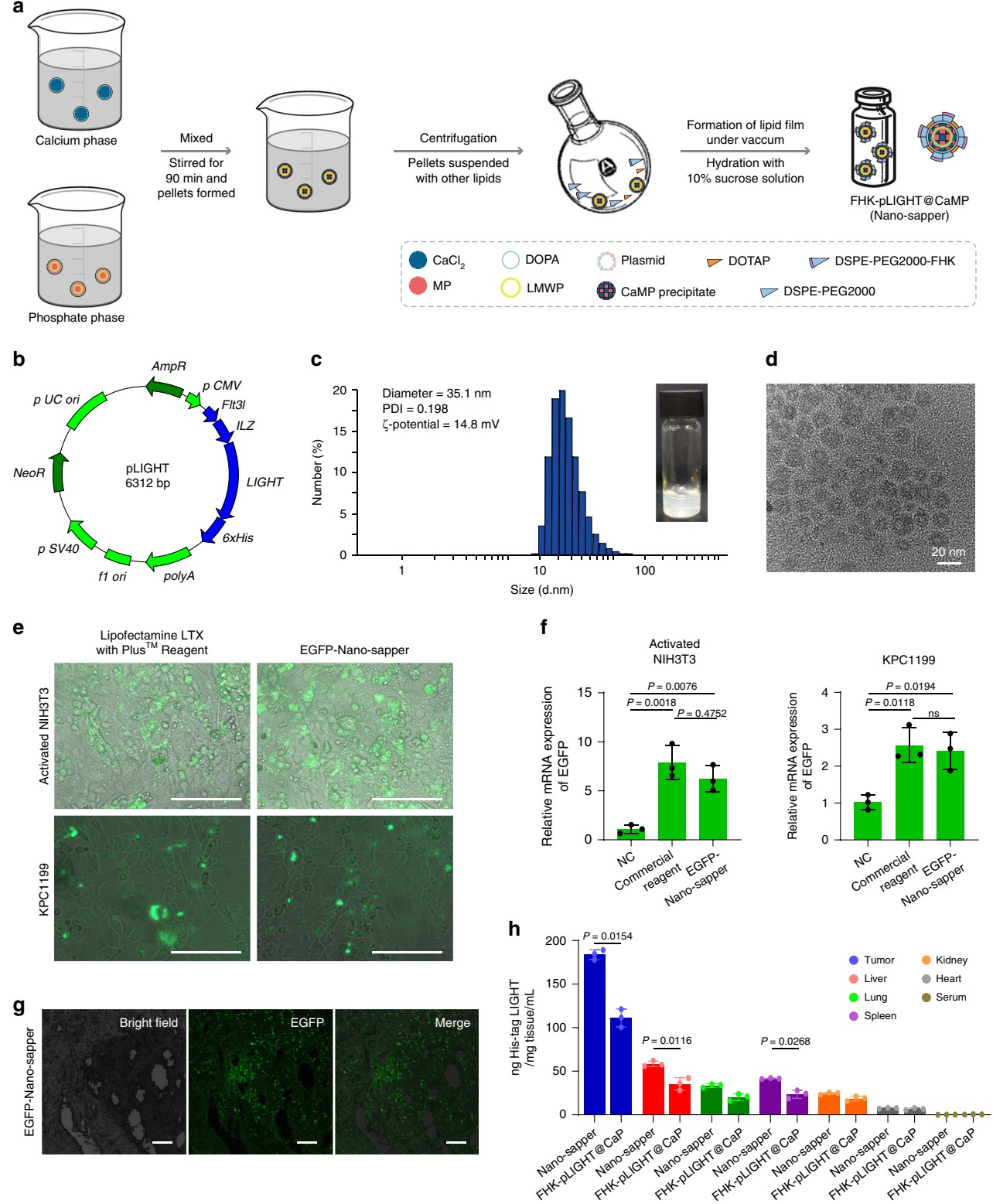

the same process except the variation in core components. For pLIGHT, the coding sequences of the extracellular domain of LIGHT (59-239 aa) and the C-terminal trimerization domain were incorporated to assemble LIGHT plasmid (Fig. 2b). The diameter of Nano-sapper was ~35 nm or 20 nm as determined by dynamic light scattering (DLS) analysis or transmission electron microscopy (TEM) (Fig. 2c, d and Supplementary Table 1). The surface charge of Nano-sapper was around 15 mV. The

encapsulation efficiency (EE) of plasmid was 51.4% at the optimized feeding we screened as quantified by Hoechst 33258 (Supplementary Fig. 11). MP was encapsulated in Nano-sapper with a relatively high efficiency (EE = 53.2 ± 2.1%), and still could be converted to α-M after incubation with either activated NIH3T3 or KPC1199 cells (Supplementary Fig. 12a and b). Besides, Nano-sapper has not shown obvious cytotoxicity to activated NIH3T3 cells, and the slightly increased cytotoxicity

**Fig. 2 Preparation and characterization of Nano-sapper. a** Nano-sapper was prepared via reversed-phase microemulsion followed by thin-film hydration. MP, α-mangostin phosphate; LMWP, low molecular weight protamine. **b** Schematic representation of plasmid encoding 6 × His tag fused LIGHT. **c, d** Visual appearance and size distribution of Nano-sapper were detected by DLS and TEM. The experiments were repeated twice independently. **e** Representative images of the transfection efficiency of EGFP coding Nano-sapper in activated NIH3T3 and KPC1199 cells. Scale bars, 100 μm. The experiments were repeated three times independently. **f** Relative mRNA expression of EGFP in activated NIH3T3 and KPC1199 cells. NC, negative control, the non-EGFP coding vector plasmid-loaded nanoparticle (FHK-pVector@CaMP), positive control was the commercial reagent, lipofectamine LTX with PLUS[TM] Reagent ($n = 3$ biologically independent samples). **g** The expression of EGFP in tumor. Scale bars, 100 μm. The experiments were repeated three times independently. **h** The expressions of 6 × His-tagged LIGHT in different organs were quantified by ELISA ($n = 3$ mice). Data are presented as mean ± s.d. One-way ANOVA with Bonferroni multiple comparisons post-test was used for (**f**) and two-tailed unpaired Student's *t*-test was used for (**h**). ns, not significant. Error bars represent s.d. Source data of (**h**) are provided as a Source Data file.

of Nano-sapper ($IC_{50} = 76.7$ μM) compared with FHK-CaMP ($IC_{50} = 88.5$ μM) was possibly due to the LIGHT (Supplementary Fig. 12c and d). The transfection efficiency of Nano-sapper was assessed with the assistance of an additionally constructed plasmid which could simultaneously encoding LIGHT and EGFP (Supplementary Fig. 13a). Comparing with commercially available Lipofectamine LTX with Plus™ Reagent, EGFP-Nano-sapper exhibited similar transfection efficiency in both activated NIH3T3 (10 ng/mL TGF-β pretreated) and KPC1199 cells (Fig. 2e, f and Supplementary Fig. 13b). We further employed immunohistochemistry (IHC) and enzyme-linked immunosorbent assay (ELISA) to test whether Nano-sapper was capable of transfecting and secreting LIGHT in vivo. EGFP-Nano-sapper or Nano-sapper was i.v. administered twice a week for 1 week to C57BL/6 mice orthotopically bearing KPC1199, and EGFP or LIGHT was found in tumor 1 day after the second dose (Fig. 2g, h). The concentration of LIGHT following the Nano-sapper treatment was nearly twice as high as that of FHK-pLIGHT@CaP. These results indicated that Nano-sapper was successfully prepared and able to transfect and express the encoded LIGHT.

**The determination of PDAC targeting ability of FHK peptide.**
Next, we evaluated the homing ability of Nano-sapper. Tenascin-C is widely expressed in the stroma of PDAC and interacts with other ECM components (e.g., fibronectin, type I collagen), and is becoming an attractive target for PDAC therapy[41]. Therefore, the FHK peptide, which possess high binding affinity to tenascin C, was proposed as a suitable targeting ligand for Nano-sapper. To determine the PDAC targeting efficiency, DiR-labeled Nano-sapper was i.v. injected in orthotopic KPC1199 model. As shown in Supplementary Fig. 14, tumor slices from mice receiving Nano-sapper showed stronger fluorescence, about 1.76-fold as much as that of non-FHK decorated nanoparticle, indicating that FHK peptide could facilitate the tumor accumulation of Nano-sapper.

**Nano-sapper attenuated the stromal barrier in PDAC.** To investigate the physical obstacles reduction in IET, we first focused on the effects of Nano-sapper on stroma attenuation. The optimized in vivo dosage of MP with α-SMA and collagen as the determinants in orthotopic KPC1199 model was initially investigated. Results showed that when MP reached 13.9 mg/kg, the utmost decrease in α-SMA and collagen were achieved (Supplementary Fig. 15). Therefore, this dosage was used for the following experiments. As shown in Fig. 3a, after four consecutive dosing regimens, tumors were subjected to IHC analysis, Masson's trichrome stain and western blotting to measure α-SMA, FAP, fibronectin and collagen. All the MP contained treatments (FHK-CaMP and Nano-sapper) significantly downregulated α-SMA, FAP, fibronectin and collagen content, signified the deactivation of CAFs (Fig. 3b). Nano-sapper achieved a reduction of 76% in α-SMA, 64% in FAP, 28% in fibronectin, and 85% in collagen compared with saline (Fig. 3c, d). Although it has been

reported that LIGHT might promote fibrosis in skin and lung synergized with TGF-β by increasing α-SMA and collagen expression[42], we did not observe visible impacts of FHK-pLIGHT@CaP on the regulation of CAFs and collagen deposition. Consistently, as shown in Fig. 3e, f, expression of α-SMA, FAP and fibronectin within tumors were significantly decreased after the Nano-sapper treatment. Collectively, these results demonstrated that Nano-sapper could attenuate the stromal barrier within PDAC.

**Nano-sapper normalized the tumor blood vessels PDAC.** To determine the impacts Nano-sapper on the structural and functional features of vessels, the tumor sections were further stained with anti-CD31, anti-VE-cadherin and anti-ICAM-1. As shown in Fig. 4a, b, following the treatment of Nano-sapper, there were no significant differences among the groups in the abundance of CD31 positive tumor blood vessels (as indicated by white arrows). Tumors after Nano-sapper treatment exhibited the typical features of normalized tumor vessels including length reduction and decreased structural heterogeneity which were routinely observed in untreated tumors[43]. The integrity of VE-cadherin-constituted adhesive junction is essential for cross-endothelium permeability. Nano-sapper induced the highest expression of VE-cadherin. Furthermore, ICAM-1, another determinant of *trans*-endothelial migration of lymphocytes, was also elevated in tumors after the FHK-pLIGHT@CaP and Nano-sapper treatments (Fig. 4c, d). Compared with FHK-CaMP and FHK-pLIGHT@CaP, Nano-sapper achieved the highest extent of vessel normalization, which could be due to the synergistic effects of MP and LIGHT.

**Nano-sapper exhibited self-enhanced accumulation in PDAC.** We proposed that the attenuated stroma and normalized vessel would make PDAC available for the accumulation of nanoparticles or macromolecules. To test our hypothesis, we labeled the CaP liposome (~30 nm) with DiR to investigate the regulated TME. At day 0, KPC1199 cells ($1 \times 10^6$) were orthotopically inoculated. At days 11, 14, and 18, three randomly selected mice in each group were intravenously injected DiR-labeled CaP and imaged by IVIS™ (Fig. 5a). As shown in Fig. 5b, c, before the treatment started, the initial accumulation in tumors among different groups were similar. After two consecutive doses, CaP achieved the highest accumulation in Nano-sapper treated group (at day 14), which was further enhanced by another two doses (at Day 18). DiR signal of Nano-sapper treated group was 1.79- and 1.67-fold higher than that of the FHK-CaMP and FHK-pLIGHT@CaP treatment, respectively (Fig. 5d) (at day 18). There are two reasons we selected CaP here. First, the similar size of CaP makes it suitable for mimicking other biomacromolecules such as IgG[44]. Second, it would help us to deeply understand the in vivo fate of Nano-sapper. As the stromal barrier has been weakened by MP, the decreased interstitial pressure would allow more therapeutics to penetrate the core of tumor. Besides, the

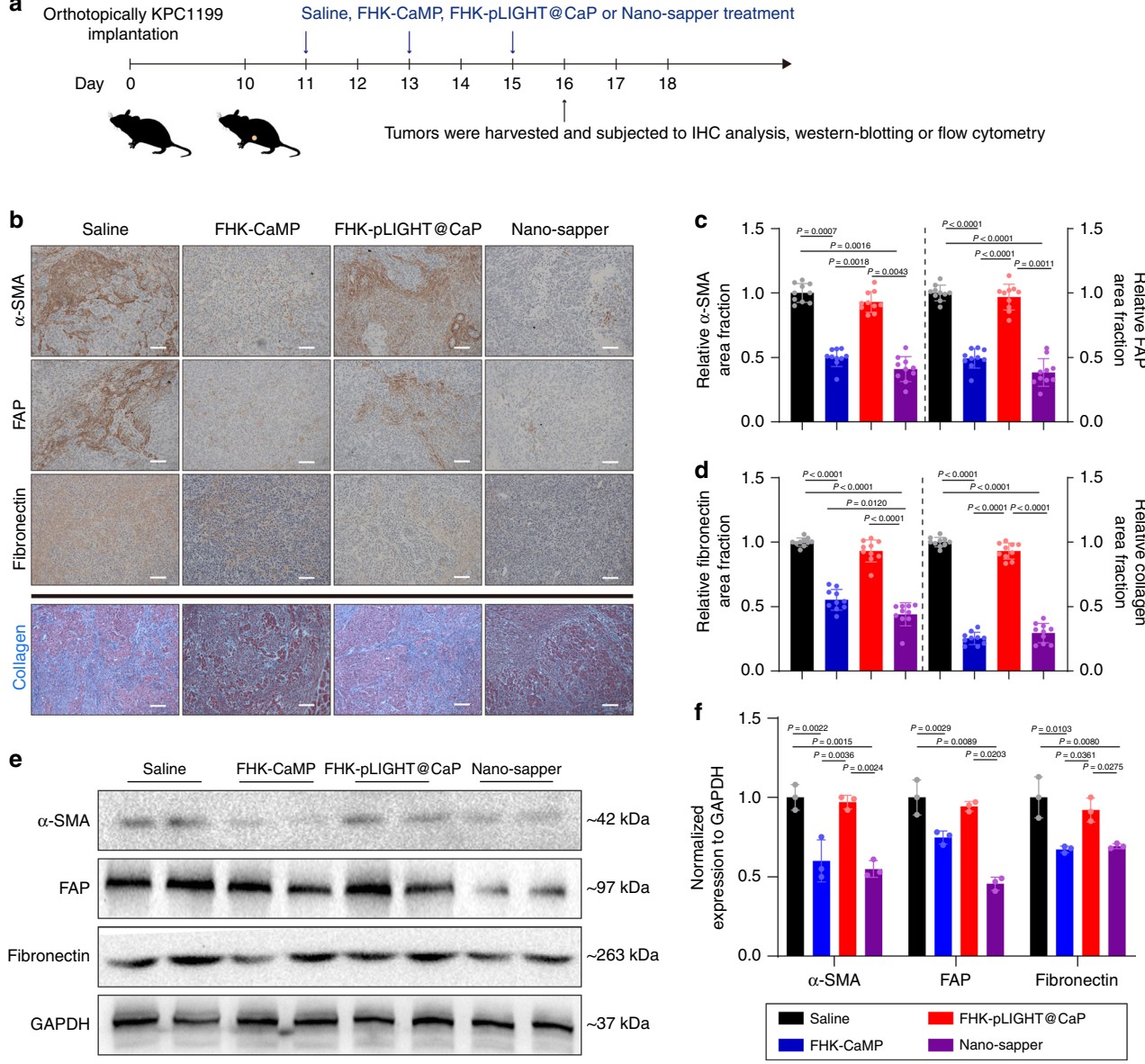

**Fig. 3 Nano-sapper reversed CAFs activation and reduced stromal composition in PDAC. a** Six-week-old male C57BL/6 mice were orthotopically inoculated with KPC1199 cells (1 × 10⁶) on day 0. Saline, FHK-CaMP (MP = 13.9 mg/kg), FHK-pLIGHT@CaP (25 μg plasmid per mouse) and Nano-sapper (MP = 13.9 mg/kg, 25 μg plasmid per mouse) were i.v. administered on day 11, 13, and 15. On day 16, tumors were harvested for analysis. **b–d** Tumor slices were stained for α-SMA, FAP, fibronectin and collagen, and analyzed by ImageJ. Scale bars, 100 μm (n = 10 section images from three mice). Data were pooled from three independent experiments. **e, f** Tumors were lysed to detect expression of α-SMA, FAP and fibronectin (n = 3 mice). Data were pooled from three independent experiments. Data are presented as mean ± s.d. One-way ANOVA with Bonferroni multiple comparisons post-test was used for (**c**), (**d**), and (**f**). Error bars represent s.d. Source data of (**e**) are provided as a Source Data file.

extensive spatial distribution of vessels induced by LIGHT, could in turn facilitated CaP distribution. We also observed the accumulation increment rate of CaP from Nano-sapper treated tumors slightly decreased from days 14 to 18 (Fig. 5c). It might be the recovery of tight conjugation between endothelium that reduced the leakage of CaP. These results again indicated a combination effects of MP and LIGHT in reducing physical obstacles.

**Nano-sapper facilitated the infiltration of CTLs in PDAC.** We continued to determine the effects of Nano-sapper on lymphocytes infiltration through flow cytometry and IHC. It was noteworthy that clusters of high endothelial venules (HEVs, marked by MECA79) were found in those tumors receiving the

treatments containing LIGHT (FHK-pLIGHT@CaP and Nano-sapper) (Fig. 6a, b). Generally, HEVs express peripheral node addressin and facilitate lymphocyte trafficking into the secondary lymphoid organs such as lymph nodes and spleen. The emergence of HEVs in tumor indicated a potential enhanced lymphocyte infiltration and spontaneous antitumor immunity. By collecting single-cell suspensions from orthotopically implanted KPC1199 tumors, we found that Nano-sapper significantly increased the ratio of CD45+CD3+CD8+ T cells, CD45+CD3+ CD4+ T cells and CD45+B220+ B cells, but decreased the ratio of Tregs and F4/80+ macrophages (Fig. 6c, Supplementary Figs. 16 and 17). Furthermore, the elevated CD8+ T/CD4+ T and CD4+ T/Treg indexes (Fig. 6d and Supplementary Fig. 18a), as indicators that positively correlated with improved patients' survival,

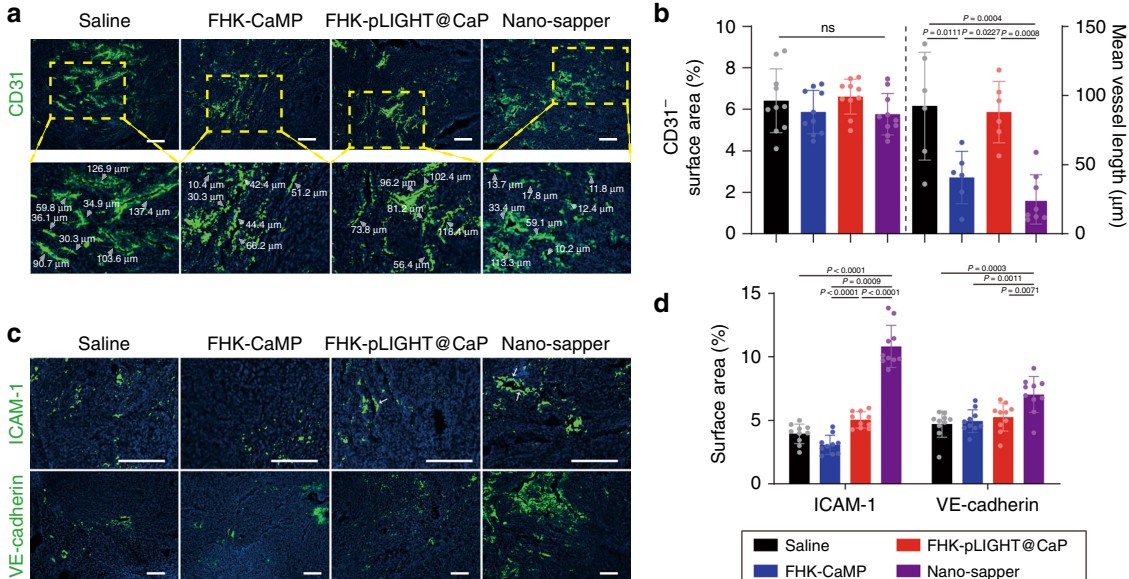

**Fig. 4 Nano-sapper normalized tumor blood vessels and increased cell adhesion molecules in PDAC. a**, **b** Tumor slices were stained with anti-CD31 and analyzed by ImageJ. Scale bars, 100 μm. $n = 10$ section images from three mice. Data were pooled from three independent experiments. Grey arrows pointed the represented vessels. **c**, **d** Tumor slices were stained with anti-VE-cadherin and anti-ICAM-1 for analysis. Scale bars, 100 μm. $n = 10$ section images from three mice. Data were pooled from three independent experiments. Cell nuclei was stained by DAPI. Data are presented as mean ± s.d. One-way ANOVA with Bonferroni multiple comparisons post-test was used for (**b**) and (**d**). ns, not significant. Error bars represent s.d.

were observed in Nano-sapper treated tumors. The down-regulation of Tregs was possibly due to the LIGHT secretion which has been reported of reversing Tregs-mediated immune suppression[27]. Since the location of CTLs determines the initiation of immune response, we further investigated the spatial distribution of CD8[+] T cells via IHC analysis. In the untreated PDAC group (Fig. 6e), we found that CD8[+] T cells were mostly scattered along the margins of tumor. In contrast, following the Nano-sapper treatment, CD8[+] T cells were found both in the margins and core of the tumor, forming clusters of T cells that resembled the morphology features of secondary lymphoid organizations (SLOs). These phenomena drove us to further investigate the spatial distribution of different lymphocytes components. As shown in Fig. 6f and Supplementary Fig. 18b, Nano-sapper treatment induced organized intratumoral lymphocyte clusters that closely resembled SLOs. Histological analysis showed that these structures harbored distinct lymphocyte subpopulation, the central dense T cell zones and the surrounding B cell zones. We also observed that the cells in the rim of T cell zones displayed B220 signal, a marker for T cell activation[45], which suggested a possible bystander effects of T cells[46]. Given the presence of HEVs in PDAC from Nano-sapper treatment, we believed that Nano-sapper induced intratumoral TLSs, which have been considered as the sites where adaptive immune response were generated to attack local tissues and have been documented in various human tumor types as an indicator of favorable patients' outcome[47]. We also determined the levels of those chemoattractants that correlated with recruitment of T or B lymphocytes, and found that both CCL21 and CXCL13 were significantly upregulated in PDAC following the Nano-sapper treatment (Fig. 6g, h). Besides, since the incorporated plasmids might sensitize TME through the cGAS-STING signaling pathway, we additionally prepared the nanoparticle loaded with vector plasmid containing scrambled LIGHT sequences (FHK-pVector@CaMP) and treated the KPC1199-bearing mice with the same schedule. As shown in Supplementary Fig. 19, neither the signal of CD8 (IHC) nor the number of CD45[+]CD3[+]CD8[+] T cells was enhanced following the FHK-pVector@CaMP

treatment, indicating that the nonsense plasmid did not induce observable changes in the infiltration of T cells. After all, these observations corroborated our previous results, confirmed the ability of combined infiltration enhancement of CTLs by Nano-sapper, and implied the necessity of combination with α-PD-1 in PDAC immunotherapy.

**Nano-sapper synergizes with α-PD-1 therapy in PDAC models.** The antitumor immune response is a complex process which affected by the location and activation status of CTLs[48]. As Nano-sapper has exhibited profound effects on physical obstacles reduction and recruiting signal stimulation, we combined α-PD-1 to further evaluate the therapeutic effects in syngeneic KPC1199 orthotopic model and Panc02 orthotopic model. As indicated in Figs. 7a and 8a, mice were treated with saline, gemcitabine (Gem), α-PD-1, Nano-sapper and Nano-sapper + α-PD-1. The combination of Nano-sapper and α-PD-1 significantly prolonged animal survival compared with other groups (Fig. 7b and Fig. 8b, Supplementary Tables 2 and 3). α-PD-1 exhibited no obvious effects, while the combination of Nano-sapper and α-PD-1 achieved the most profound tumor inhibition (Figs. 7c, d and 8c, d). Besides, Nano-sapper combined with α-PD-1 induced the largest tumor necrosis (Figs. 7e, 8e).

Furthermore, the tumor metastasis was monitored one day after the end of therapy. Morphological and histological analysis (Fig. 7f, g) showed that large nodules of metastasis (indicated by white arrows) and focal necrosis (indicated by yellow arrows) presented in the liver and kidney from saline group. Other monotherapies also showed suppression of liver metastasis while the combination of Nano-sapper with α-PD-1 was able to significantly inhibit or even abrogate metastasis.

**Toxicity evaluation.** Safety of Nano-sapper was further investigated. C57BL/6 mice orthotopically bearing KPC1199 or healthy mice were administered with saline, Gem, α-PD-1, Nano-sapper and Nano-sapper + α-PD-1, respectively, according to therapeutic dosing regimen. There were not any noticeable

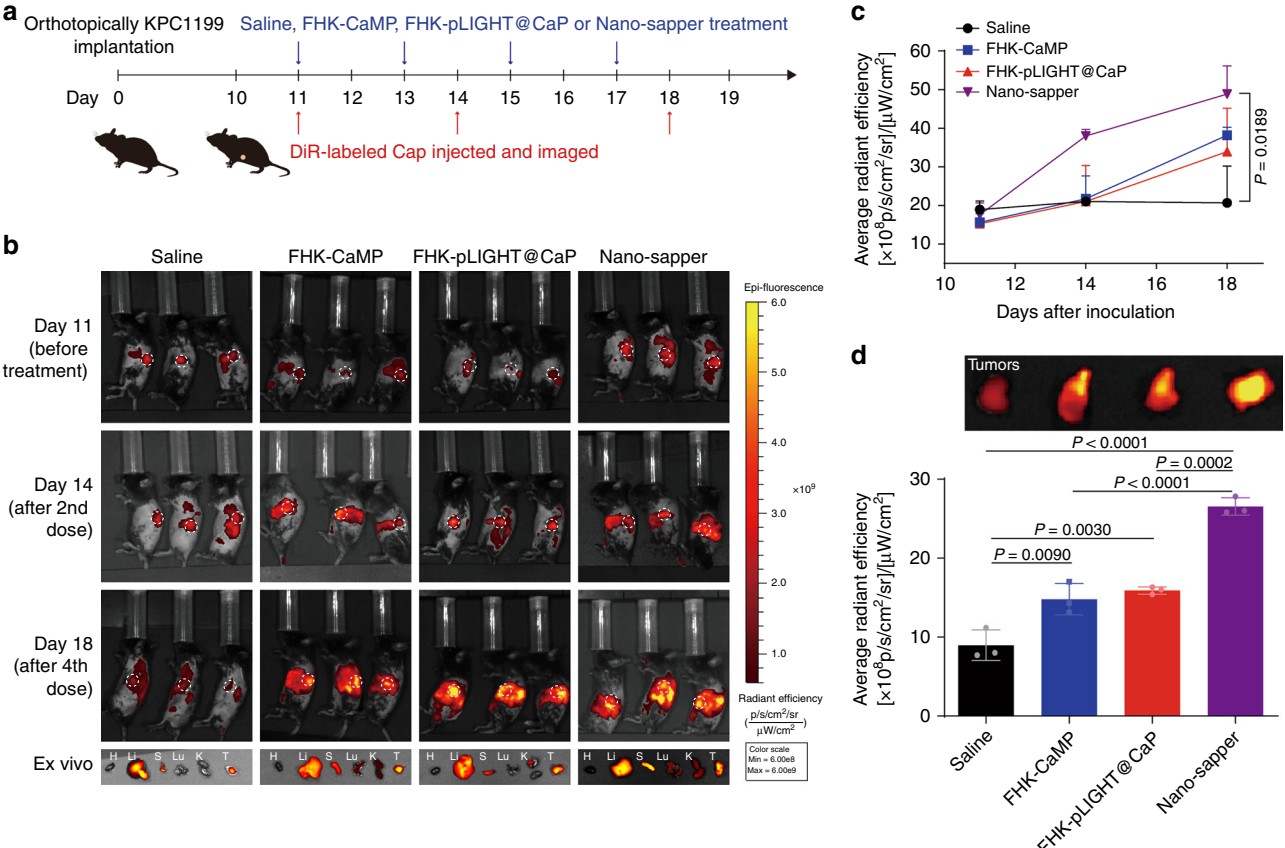

**Fig. 5 Nano-sapper exhibited self-enhanced accumulation in PDAC. a** Six-week-old male C57BL/6 mice were orthotopically inoculated with KPC1199 cells (1 × 10⁶) on day 0. Saline, FHK-CaMP (MP = 13.9 mg/kg), FHK-pLIGHT@CaP (25 μg plasmid per mouse) and Nano-sapper (MP = 13.9 mg/kg, 25 μg plasmid per mouse) were i.v. administered on day 11, 13, 15, and 17, and DiR-labeled CaP was i.v. administered on day 11, 14, and 18 (DiR = 1 mg/kg), and subjected to IVIS™ for in vivo imaging. On day 18, tumors were harvested for ex vivo detection. **b, c** DiR signal was monitored to reflect the accumulation of CaP. H, heart; Li, liver; S, spleen; Lu, lung; K, kidney (*n* = 3 mice). **d** Tumors were extracted to detect DiR signal (*n* = 3 mice). Data are presented as mean ± s.d. One-way ANOVA with Bonferroni multiple comparisons post-test was used for (**c**) and (**d**). Error bars represent s.d. Source data of (**c**) are provided as a Source Data file.

morphological changes in the heart, spleen, and lungs (Fig. 9a). The body weight of those mice given with the Gem, α-PD-1 or Nano-sapper treatment exhibited slow decline, while that of those animals receiving the combination therapy of Nano-sapper and α-PD-1 held steady since the beginning of treatment (Fig. 9b). We also collected peripheral blood for hematological and serum biochemical analyses. The saline treatment induced elevation in alanine aminotransferase (ALT), aspartate aminotransferase (AST), and blood urea nitrogen (BUN), which demonstrate severe liver damage or slightly kidney toxicity caused by tumor progression. Nano-sapper + α-PD-1 did not induce significant variation in ALT, AST, and BUN (Fig. 9c). All these results indicated acceptable biocompatibility of Nano-sapper after combined with α-PD-1.

## Discussion

ICB have shown great promise for the treatment of various advanced tumor types but failed in IET. The major challenges are the dense stroma, defective vasculatures and the lack of signals recruiting CTLs, which collectively lead to the void of CTLs, and thus leave us an unmet need of developing innovative combination therapy. As a proof of concept, Nano-sapper that loaded with antifibrotic MP and plasmid incorporating immune-enhanced LIGHT was tested in murine PDAC. α-M could reverse the activation of CAFs and decrease the stromal deposition, and LIGHT could restore the vasculature and stimulating recruiting

CTLs signals. In an alternative perspective, the anti-fibrosis therapy just also coincidently facilitates the extravasation of LIGHT, and thus broaden the engagement between α-PD-1 and tumor cells. As a result, the combination of checkpoint inhibitors with Nano-sapper should enhance its efficacy.

Utilizing pro-inflammatory cytokine expressing plasmid to regulate the immune-suppressive TME is of great attraction, but it is possible that the loaded plasmid also sensitizes the TME through cGAS-STING pathway. For Nano-sapper, this possibility was excluded by incorporating a scrambled LIGHT sequences coded plasmid (Supplementary Fig. 19). It was found that the nonsense plasmid did not induce observable changes in T cell infiltration. There are two possible explanations for the absence of cGAS-STING mediated immune-sensitizing. First, the compaction by protamine reduces the exposure of plasmid to cGAS. Protamine sheltered the pentose-phosphate backbone of plasmid through the interaction between positively charged amino acid residues of protamine and negatively charged phosphate groups of plasmids[49]. Since the backbone of double-stranded DNA happens to be the binding site of cGAS[50], the activation of cGAS-STING could be relieved by protamine. Second, Second, STING activation could possibly be counteracted by α-M. For example, α-M have been reported to decrease the activation of several signaling pathways including IL-1 and IL-6, which are inflammatory cytokines driven by cGAS-STING activation[51].

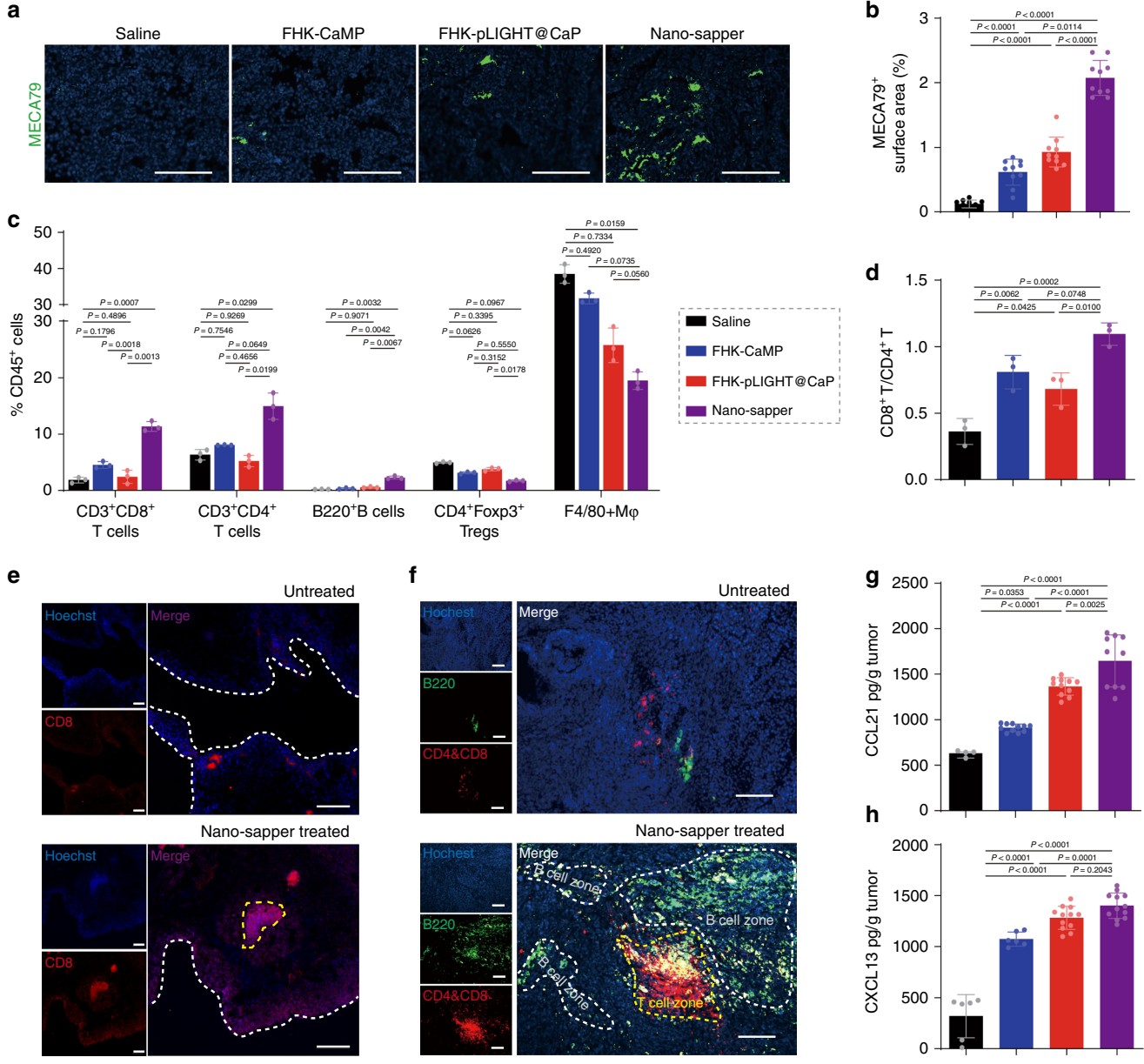

**Fig. 6 Nano-sapper enhanced CTLs infiltration and initiated the TLSs formation in TME. a**, **b** Tumor slices were stained for MECA79 and analyzed by ImageJ. Scale bars, 100 μm. $n = 10$ section images from three mice. The experiments were repeated three times independently. **c**, **d** Flow cytometry analysis of immune cells within TME ($n = 3$ mice). **e**, **f** Nano-sapper treatment promoted T cell penetration into deeper region of PDAC and induced the formation of TLSs. Scale bars, 100 μm. Data were pooled from three independent experiments. **g**, **h** Chemoattractants involved in lymphocyte recruitment were elevated within TME. (For CCL21, Saline, $n = 4$ mice; FHK-CaMP, $n = 11$ mice; FHK-pLIGHT@CaP, $n = 11$ mice; Nano-sapper, $n = 10$ mice. For CXCL13, Saline, $n = 6$ mice; FHK-CaMP, $n = 6$ mice; FHK-pLIGHT@CaP, $n = 12$ mice; Nano-sapper, $n = 12$ mice.) Data are presented as mean ± s.d. One-way ANOVA with Bonferroni multiple comparisons post-test was used for (**b**), (**c**), (**d**), (**g**), and (**h**). Error bars represent s.d.

Since many studies have reported the drugs show promises in pre-clinical tumor models did not always benefit the clinic, model selection is an important concern in this work. Through literature search and our results, KPC1199 orthotopic model can recapitulate the clinical features of the cognate human conditions. (1) Its similar histopathological features with human PDAC, such as the duct-like structures, dense stroma, the compressed vessels and the abnormally activated fibroblasts (Supplementary Fig. 20a). (2) Its shared genomic features with human PDAC. Primary tumor cell line KPC1199 derived from KPC pancreatic ductal adenocarcinoma mouse (LSL-*Kras*[G12D/+]; LSL-*Trp53*[R172H/+]; *Pdx-1-Cre*, on C57BL/6 background) possess *Kras* and *p53* double mutation, which reflect the most prevalent

genotype of PDAC in clinic[52]. (3) Its similar inflammatory features with human PDAC development. The interleukin 6 (IL-6), which is responsible for the progression from chronic pancreatitis to pancreatic cancer[53], is extensively expressed in the KPC1199 orthotopic model. (4) Its similar therapeutic features with human PDAC, such as the metastasis along with the primary tumor and resistance to gemcitabine[54]. Considering the genetic diversity of PDAC in clinic, We also tested Nano-sapper in Panc02 transplantation model which has both *Kras* and *Smad4* (occurred in ~50% of pancreatic cancers) mutations[55–57]. The prolonged survival by Nano-sapper combined with α-PD-1 implying its potential in treating PDAC with heterogeneous genotypes.

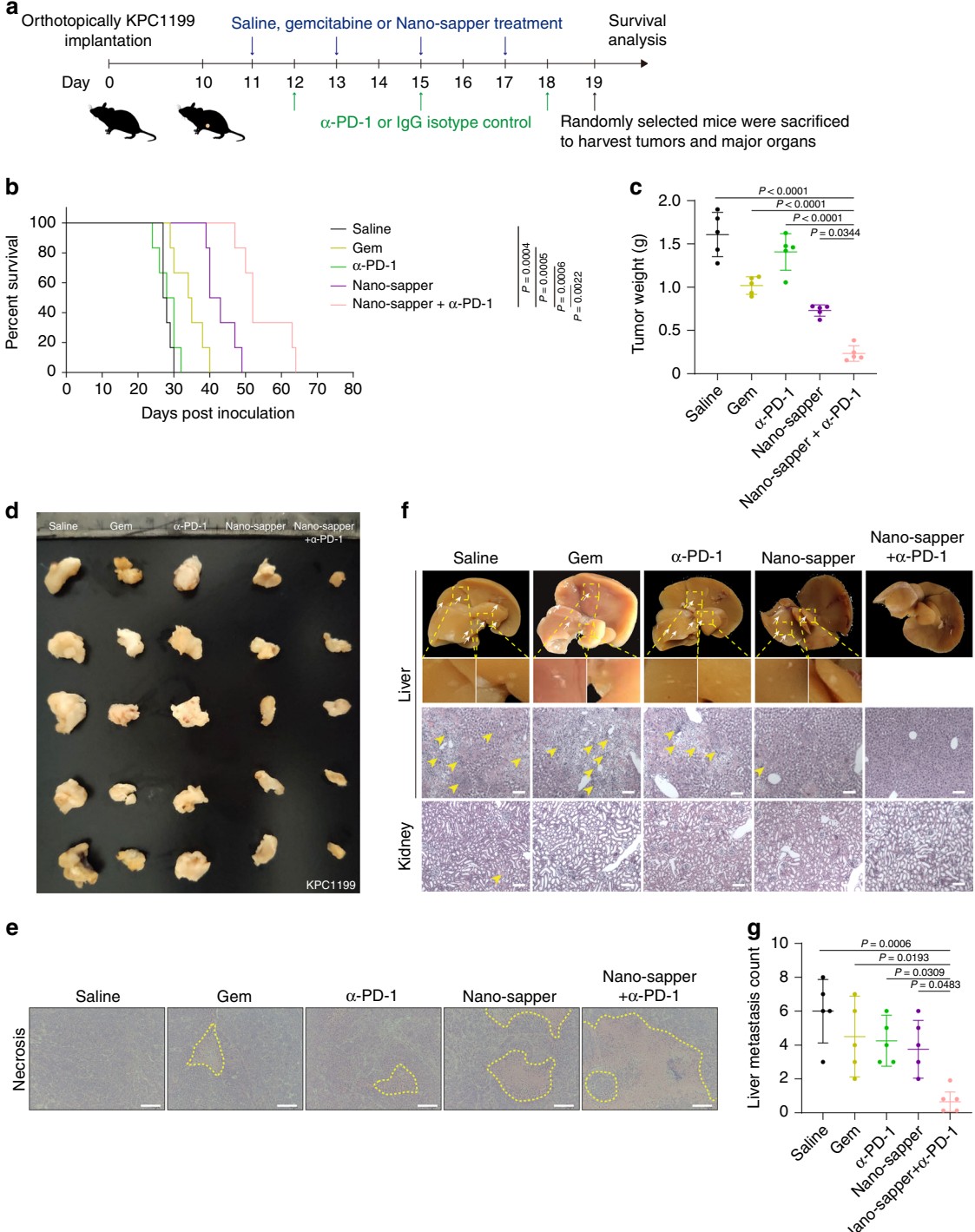

**Fig. 7 Nano-sapper synergized with α-PD-1 to suppress tumor growth and metastasis in KPC1199 PDAC model. a** Six-week-old male C57BL/6 mice were orthotopically inoculated with KPC1199 cells ($1 \times 10^6$) on day 0. Saline, Gem (15 mg/kg) and Nano-sapper (MP = 13.9 mg/kg, 25 μg plasmid per mouse) were i.v. administered every other day and α-PD-1 or IgG isotype (200 μg per mouse) were i.p. administered every 2 days ($n = 12$ mice). One day after the final α-PD-1 treatment, five mice were randomly sacrificed to extract tumors and major organs, while the rest mice were continually subjected to survival analysis. **b** Mice survival curves in each treatment group ($n = 6$ mice). **c, d** Tumor inhibition of different treatments ($n = 5$ mice). **e** H&E staining histology of tumor necrosis. Scale bars, 100 μm. **f, g** H&E staining histology of tumor metastasis in liver and kidney. Nodules of metastasis were indicated by white arrows and focal necrosis were indicated by yellow arrows ($n = 5$ mice). Data are presented as mean ± s.d. One-way ANOVA with Bonferroni multiple comparisons post-test was used for (**c**) and (**g**). Kaplan–Meier analysis with log-rank Mantel-Cox test (two-sided) was used for (**b**). Error bars represent s.d.

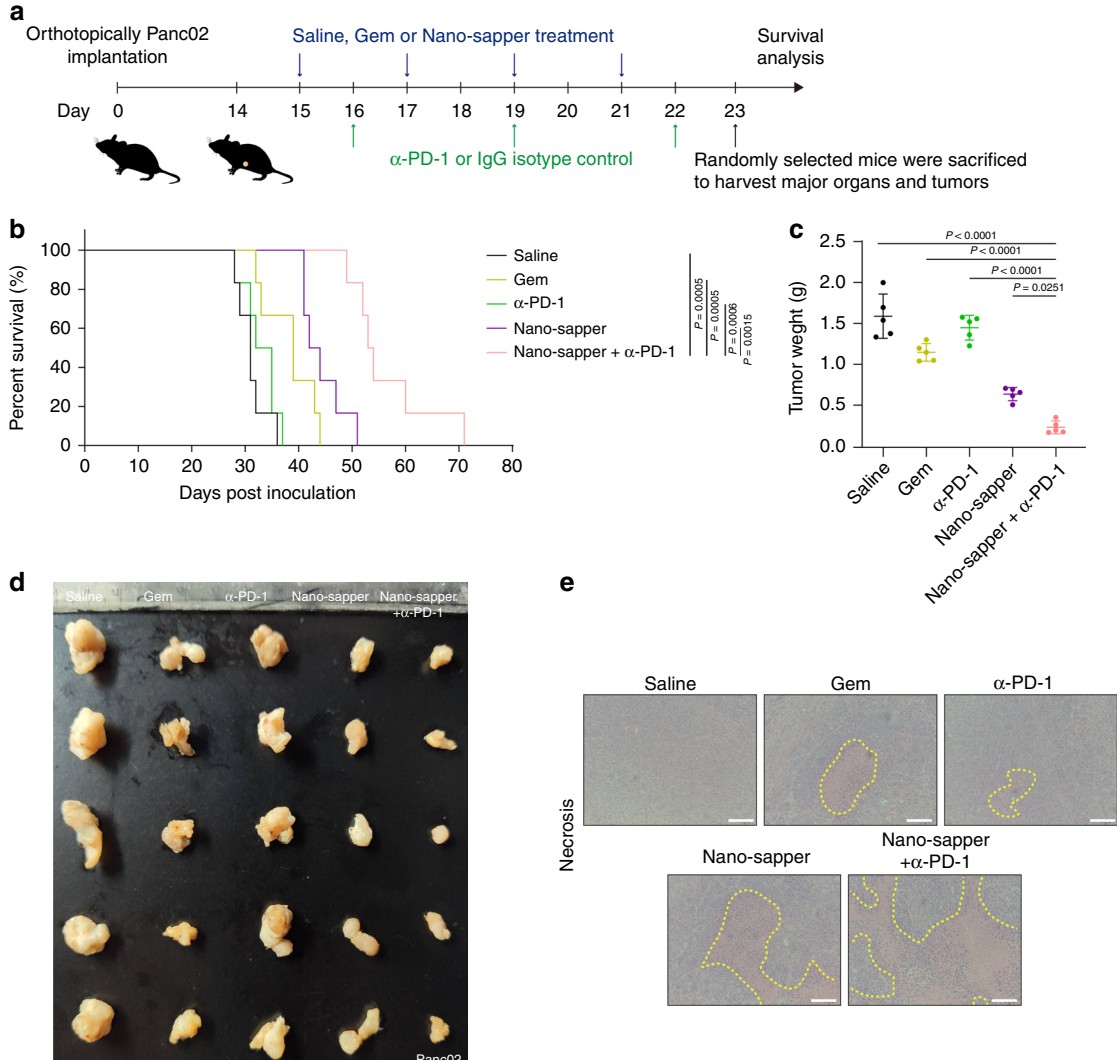

**Fig. 8 Nano-sapper synergized with α-PD-1 to prolong the survival of Panc02 PDAC-bearing mice. a** Six-week-old male C57BL/6 mice were orthotopically inoculated with Panc02 (1 × 10⁶) on day 0. Saline, Gem (15 mg/kg) and Nano-sapper (MP = 13.9 mg/kg, 25 μg plasmid per mouse) were i.v. administered every other day and α-PD-1 or IgG isotype (200 μg per mouse) were i.p. administered every 2 days (n = 12 mice). One day after the final α-PD-1 treatment, five mice were randomly sacrificed to extract tumors and major organs, **b** Mice survival curves in each treatment group (n = 6 mice). **c**, **d** Tumor inhibition of different treatments (n = 5 mice). **e** H&E staining histology of tumor necrosis. Scale bars, 100 μm. Data are presented as mean ± s.d. One-way ANOVA with Bonferroni multiple comparisons post-test was used for (**c**). Kaplan–Meier analysis with log-rank Mantel-Cox test (two-sided) was used for **b**. Error bars represent s.d.

Immune toxicity associated with immunotherapy can cause damages to a variety of organ systems. In this work, there were two potential sources of the toxicity. On one hand, as a member of the TNF superfamily, the immune-enhanced cytokine LIGHT, would cause certain side effects after the systemic administration. On the other hand, The tumor-associated antigens were also expressed by a range of normal tissues, which might draw the attacks by CTLs[58]. Therefore, it is required to precisely express the encoded LIGHT and increase recruiting signals in tumors to minimize the immune-related adverse events (irAEs) and the off-target toxicity of CTLs. The FHK peptide decoration allowed the tumor-preferred accumulation of Nano-sapper and enabled the localized recruiting signals stimulation. Through the whole treatment, no obvious irAEs such as rash or diarrhea were observed from Nano-sapper combined with α-PD-1.

Accumulating evidence indicates that TLSs play a major role in controlling tumor development and has been gradually recognized as a favorable prognosis on patients' survival in many advanced cancers[59]. It also has been reported the induction of TLSs could increase the antitumor immune response. Besides, the rich stromal cells filled in PDAC provide an ideal place for the initiation of TLSs. In our work, TLSs in KPC1199 model were observed from Nano-sapper treatment, which indicated the necessity of simultaneously reducing physical obstacles and stimulating recruiting signals in promoting TLSs formation.

In this study, we found that the combination of α-M and LIGHT could reduce physical obstacles and stimulate recruiting signals, which synergistically facilitated CTLs infiltration, initiated TLSs, and finally improved the potent of ICB against PDAC. The nanotechnology derived from Nano-sapper might become a general solution for targeted co-delivery of insoluble hydroxy-rich polyphenols and pro-inflammatory cytokines. Furthermore, the combination of anti-fibrosis treatment and immune-enhanced cytokine treatment would provide an effective and safe option in promoting immunotherapies, such as CAR-T and vaccines, against IET.

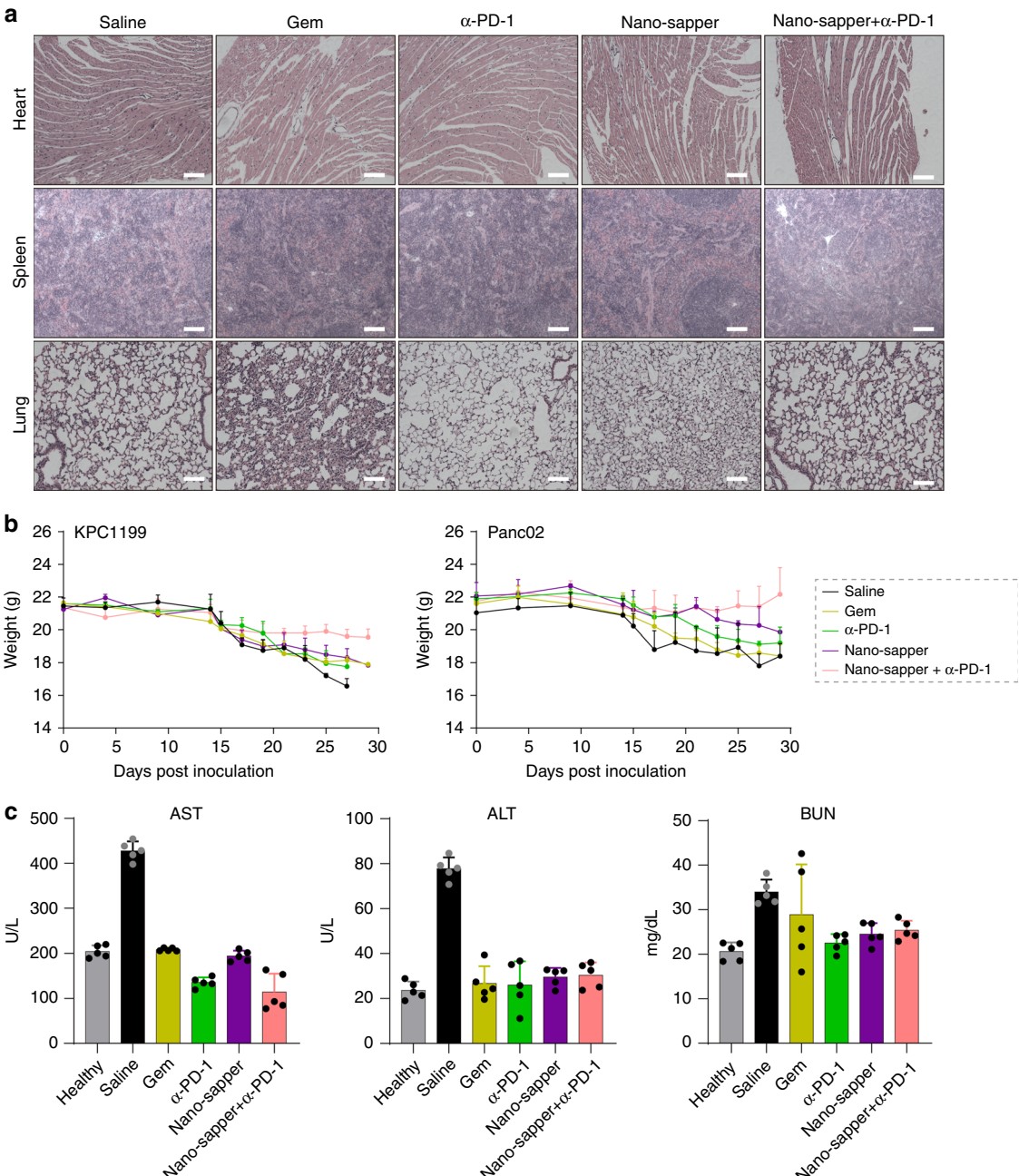

**Fig. 9 Biosafety evaluation of different treatments. a** H&E staining histology of heart, spleen and lung to evaluate the safety of different treatments. Scale bars, 100 μm. Data were pooled from three independent experiments. **b** The body weight of mice monitored during the therapy ($n = 6$ mice). **c** Systemic cytotoxicity evaluation of different treatments ($n = 5$ mice). ALT, serum alanine aminotransferase; AST, aspartate aminotransferase; BUN, blood urea nitrogen. Data are presented as mean ± s.d. Error bars represent s.d. Source data of (**b**) and (**c**) are provided as a Source Data file.

## Methods

**Materials**. (2,3-dioleoyloxy-propyl)-trimethylammonium (DOTAP) and 1,2-dio-leoyl-sn-glycero-3-phosphate (DOPA) were obtained from Avanti Polar Lipids (Alabaster, AL, USA). 1,2-distearoyl-sn-glycero-3-phosphoethanolamine-N-[amino(polyethylene glycol)-2000] (ammonium salt) (DSPE-PEG2000) was obtained from Xi'an Ruixi biotech (Xi'an, China). FHK peptide (FHKHKSPALSPV) was synthesized by China Peptides Co., Ltd (Shanghai, China). Low molecular weight protamine (LMWP), IGEPAL® CO-520, cholesterol, 1,1′-dioctadecyl-3,3,3′,3′-tetramethylindotricarbocyanine iodide (DiR), 3-[4,5-dime-thylthiazol-2-yl]-2,5-diphenyl tetrazolium bromide (MTT), 4,6-Diamidino-2-phe-nylindole (DAPI), Hoechst 33258 and phosphatase inhibitor cocktails were provided by Sigma-Aldrich (St. Louis, MO, USA). α-M were purchased from Biopurify Co., Ltd (Chengdu, China). Collagenase type IV (40510ES60), hyalur-onidase (20426ES60), DNase I (10608ES25) and Agar (70101ES76) were purchased from Shanghai yeasen biotech (Shanghai, China). Tryptone (LP0042) and Yeast extract (LP0021) were obtained from OXOID. Gemcitabine was obtained from Eli

Lilly. Anti-mouse PD-1 antibody (clone: RMP1-14) and IgG isotype (MPC-11, BP0086) are acquired from BioXcell. All the other chemical reagents and solvents were acquired from Sinopharm Chemical Reagent Co., Ltd (Shanghai, China) unless specified.

**Cells and animals**. The mouse embryonic fibroblast cell lines NIH3T3 were provided by Dr. Jibin Dong from Fudan University School of Pharmacy (Shanghai, China). The primary tumor cell line KPC1199 derived KPC pancreatic ductal adenocarcinoma mouse model (LSL-Kras$^{G12D/+}$; LSL-Trp53$^{R172H/+}$; Pdx-1-Cre, on C57BL/6 background) were kindly provided by Dr. Jing Xue from Renji Hos-pital (Shanghai, China). The murine pancreatic cancer cell Panc02 was kindly provided by Dr. Peng Wang from Fudan University Shanghai Cancer Center (Shanghai, China). Cells were cultured in DMEM supplemented with 10% fetal bovine serum and 1% penicillin-streptomycin solution at 37 °C and 5% $CO_2$ in a humidified atmosphere. Male C57BL/6 mice at 6-weeks old were obtained from

SLAC Animal Ltd. (Shanghai, China) and housed in the specific pathogen-free conditions at 18–23 °C and 40–60% humidity with free access to food and water. All the animal experiments were performed in accordance with the guidelines evaluated and approved by Institutional Animal Care and Use Committee (IACUC), Fudan University School of Pharmacy. All the animal experiments have been approved by IACUC, Fudan University School of Pharmacy.

**Synthesis and characterization of α-M phosphate.** (1) *Synthesis of S1.* In a round-bottom flask, α-M (308 mg, 0.75 mmol, 1 equiv) and DMAP (7 mg, 0.2 equiv) was dissolved in 15 mL THF. Triethylamine (30 equiv) and diethyl chlorophosphate (30 equiv) were added dropwise at 0 °C. The mixture was stirred at room temperature for 12 h under nitrogen atmosphere, leading to the formation of a precipitate. Aqueous hydrochloric acid (1 M, 30 mL) was added to the reaction mixture and the solution was stirred until the precipitate disappeared. The material was extracted with ethyl acetate and was washed with brine. The combined organic layers were dried over $Mg_2SO_4$. After filtration, the filtrate was concentrated under reduced pressure. The residue was purified by column chromatography on silica gel (Petroleum ether/ethyl acetate = 2/1) to provide S1 (370 mg, 72.3%). (2) *Synthesis of MP.* To a solution of S1 (134 mg, 0.2 mmol) in DCM was added TMSBr (260 μL, 10 equiv) at 0 °C. After stirring at RT for 10 h, methanol (5 mL) was added, and the solution was stirred at the same temperature for another 2 h. The mixture was concentrated under reduced pressure. The crude product was purified by silica gel column chromatography (methanol:dichloromethane = 10:1) to afford the desired product MP as a yellow solid (85 mg, 75%).

**MP conversion to α-M.** In vitro MP conversion to α-M by alkaline phosphatase (AP) was first evaluated. Two hundred micrograms of MP were mixed with 50 U of AP in 1 mL OPTIZYME AP buffer and incubated at 37 °C for 1 h. The mixture was then lyophilized. Then 500 μL 40% methanol was added to the powder and then subjected to HPLC analysis (Aglient 1260). The separation of α-M was achieved by using a Venusil phenyl column (100 mm × 2.1 mm, 5 μm) with methanol/water (containing 0.5% formic acid) 40:60 as the mobile phase with a flow rate of 0.5 mL/min at detection wavelength of 325 nm. The conversion of MP to α-M was further validated in live NIH3T3 cells and KPC1199 cells. FHK-CaMP containing 50 μg MP was incubated with NIH3T3 cells or KPC1199 cells. After 4 h incubation, the medium and cells were collected and lysed with 1% Triton X-100, and subjected to lyophilization. Then 500 μL 40% methanol was added, and α-M was detected by HPLC analysis. Next, Nano-sapper were i.v. injected to KPC1199 orthotopic PDAC-bearing mice at a MP dose of 10 mg/kg. 1 h after injection, the mice were sacrificed, and the tumors were harvested and analyzed for α-M concentration by a HPLC-MS method. Baicalein was used as an internal standard. The homogenized tumor tissue was extracted with methanol followed by nitrogen sweeping. After reconstituting with 40% methanol and centrifugation, the supernatant was used for HPLC analysis. The mobile phase is methanol/0.5% formic acid with a gradient elution (0–2 min 20% methanol, 2–10 min 20–90% methanol, 10–13 min 90% methanol, 13–17 min 20% methanol; 0.5 mL/min). The mass spectrometer was operated in the positive ion mode.

**Preparation and characterization of the Nano-sapper.** The inner cores of Nano-sapper were prepared by water-in-oil microemulsions in an oil phase containing cyclohexane/Igepal® CO-520 solution (72/28, v/v), as described previously with modification[35,60,61]. In general, 300 μL of 5 M $CaCl_2$ and 30 μL of 8 mg/mL LMWP was mixed in 20 mL oil phase with continuous stirring to form the calcium part. The phosphate part was prepared by adding 100 μL of 15 mg/mL MP and 100 μL of 1 mg/mL LIGHT plasmid in a separate 20 mL oil phase. Hundred microliters of DOPA (20 mg/mL) in chloroform was individually added to both calcium and phosphate part. After 5 min, the two oil phases were mixed and allowed mixing for another 45 min. Then 40 mL of absolute ethanol was added, and the mixture was centrifuged at 13,000 × g for 30 min to remove cyclohexane and the surfactant. After being extensively washed by ethanol for three times, the pellets were dissolved in 5 mL chloroform and stored in a glass vial at −20 °C for future use. To prepare the final Nano-sapper, 10 mg core was mixed with 300 μL of 20 mmol cholesterol, 300 μL of 20 mmol DOTAP, 200 μL of 20 mmol DSPE-PEG2000, and 100 μL of 20 mmol DSPE-PEG2000-FHK. After chloroform was evaporated, the residual lipids were suspended in 10% sucrose solution followed by sonication to form the final Nano-sapper. The DiR-labeled CaP were prepared by adding 300 μL 12.5 mmol $Na_2HPO_4$ (pH > 9) instead of MP while with 1% DiR added to the lipids. The particle size and zeta potential of Nano-sapper were determined by a Malvern Zetasizer Nano series (Westborough, MA). TEM images of Nano-sapper were acquired using a TEM (TEM-1400 Plus Electron Microscope, Leica, Germany). The drug-loading capacity and encapsulation efficiency of MP were measured using a UV-Vis spectrophotometer (Shimadzu UV-2401PC). FHK-CaMP and Nano-sapper were first lysed using a pH = 4 acetic acid buffer, and the concentration was determined using a standard curve. For the determination of plasmid, we used Hoescht 33258 nucleic acid stain to confirm DNA entrapment efficiency in FHK-pLIGHT@CaP and Nano-sapper as previously reported.

**The determination of plasmid concentration.** Hoechst 33258 stock (1 mg/mL in distilled $H_2O$) was previously prepared and sterilized by filtration through a 0.22 μm filter and stored at 4 °C in a light tight container. Working assay solution was prepared by adding 1 μL of stock solution for every 1 mL of assay buffer. FHK-pLIGHT@CaP and Nano-sapper were dissolved in lysis buffer (2 mmol EDTA and 0.05% Triton X-100 in pH 7.8 Tris buffer) at 65 °C for 10 min. The residual LMWP/plasmid complex was further dissociated by incubating with protease K at 37 °C for 1 h. After incubation, working assay solution was added, and samples were assessed via fluorescence spectrometry with a 360-nm, 20-nm bandwidth, excitation filter and a 460-nm, 20-nm bandwidth emission filter. Standard curves were generated using blank CaP along with known concentrations of LMWP/plasmid complexes. The calculated plasmid encapsulation efficiency is about 50%.

**Immunohistochemistry chemistry analysis.** Three mice of each group were anaesthetized 24 h after the final administration and heart perfused with saline and fixed with 4% paraformaldehyde. Tumors were harvested and fixed before gradient dehydrating by 15 and 30% sucrose solution, and then imbedded in tissue optimum cutting temperature (OCT)-freeze medium (Sakura, Torrance, CA, USA), frozen at −80 °C for slicing. Immunohistochemistry were conducted on OCT embedded tumor slices. Slices were permeabilized and blocking in 5% goat serum albumin at room temperature for 1 h. Then slices were stained with fluorescently labeled antibodies or primary rabbit anti-mouse laminin antibodies followed by secondary Alexa Flour 488-labeled goat anti-rabbit IgG antibodies. All antibodies were diluted after optimization. After staining nuclei with Prolong® Diamond Antifade Mountant with DAPI (ThermoFisher Scientific) for another 10 min, the sections were observed under fluorescence microscope (Leica DMI 4000B, Germany). The mean vessel length was quantified as previously described by using ImageJ to measure the length of continuous CD31 positive signal in the enlarged field of view, and then divided it by the number of vessels[29,62]. The Masson's trichrome assay was performed to detect collagen in tumor tissues. Tumor slides were stained using a Masson's trichrome kit by the UNC Tissue Procurement Core. Masson's trichrome staining of tumor sections was imaged using an Eclipse Ti-U inverted microscope (Nikon Corp, Japan) with ×10 objective. Five randomly selected microscopic fields were quantitatively analyzed using ImageJ software. For the Masson's Trichrome Staining, paraffin-embedded tumor sections were deparaffinized and rehydrated. The slides were then stained using a Masson's Trichrome Kit (Solarbio®, Beijing, China) following the manufacturer's instructions.

**Western blot analysis.** To examine the expression of α-SMA, FAP, fibronectin and (p)-Smad2/3, immunoblotting analysis was performed. NIH3T3 cells were seeded in a 6-well culture plate at the density of $10^6$ cells per well for 24 h. Then 10 ng/mL TGF-β was added to stimulate NIH3T3 into an activated state for 24 h. Then α-M with varied concentrations were added. After incubation for another 24 h, cells were washed with PBS twice and lysed with RIPA lysis buffer containing 1 mmol phenylmethylsulfonyl fluoride (PMSF) and phosphatase inhibitor cocktail (cOmplete®, Roche, Germany) for 10 min. Thereafter, the lysates were centrifuged at 12,000 × g for 10 min at 4 °C, followed by quantification of the total protein in supernatant through BCA assay (Beyotime Biotechnology, Haimen, China). The tissue samples were prepared in the same way with 50 mg tissue per tumor. Supernatants were then mixed with loading buffer and heated at 95 °C for 15 min, and 20 μg of protein from different samples were loaded into 8% SDS−PAGE to separate target proteins in an electrophoresis chamber system (Bio-Rad Laboratories, PA, USA). After separation, proteins were transferred to nitrocellulose filter membranes (NC) and blocked with 5% nonfat milk in TBS-T (Tris-HCl 50 mmol, NaCl 150 mmol, Tween-80 0.1%). The blots were then incubated with different primary antibodies (indicated in Supplementary method) against overnight at 4 °C, followed by incubation with the horseradish peroxidase (HRP)-conjugated anti-rabbit IgG (Cell Signaling Technology, 7074) at room temperature for 1 h. Finally, the bands were detected with electrochemiluminescence western-blotting substrate for imaging, and semi-quantified with ImageJ software. The uncropped and unprocessed scans of blots are presented in Source Data file.

**Flow cytometry assay.** Tumor-infiltrating immune cells were characterized by flow cytometry. In brief, mice of each group were anaesthetized 24 h after the final administration. Tumors were incubated in dissociation buffer (1 mg/mL collagenase, 1 mg/mL hyaluronidase, and 10 μg/mL DNase I) at 37 °C for 1 h to generate a single-cell suspension. For each test, $10^6$ cells were stained with fluorescently labeled antibodies for surface marker expression analysis followed by fixation with 4% paraformaldehyde. Cells were detected via FACS (BD FACSAria II) and results were analyzed via FlowJo software (Tree Star Software, San Carlos, California, USA). Fluorescence conjugated antibodies used for flow cytometry are listed in Supplementary method.

**Transfection assay.** To examine the transcription of EGFP, flow cytometry and RT-qPCR assay was performed. The activated NIH3T3 cells (10 ng/mL TGF-β pretreated) or KPC1199 were grown until 90–95% confluent in 6-well plates. Then nanoparticle loaded with plasmid (2 μg) were added to each well in the presence of Opti-MEM. The medium was refreshed 6 h post transfection and incubated for another 24 h. The non-EGFP coding vector plasmid-loaded nanoparticle (FHK-pVector@CaMP) was used as the negative control (NC) and the commercial reagent, lipofectamine LTX with PLUS™ Reagent, was used as a positive control following the manufacturer's

protocol. After the imaging, the expression of EGFP was measured via flow cytometry and RT-qPCR. The primer sequences are listed. *EGFP* (forward: 5′-AAGGGCAT CGACTTCAAGG-3′, reverse: 5′-TGCTTGTCGGCCATGATATAG-3′); *β-actin* (forward: 5′-GCACCACACCTTCTACAATG-3′, reverse: 5′-TGCTTGCTGATC CACATCTG-3′). *β-actin* was used as an internal reference gene to normalize the expression of the detected genes. Relative quantification of gene was analyzed by the comparative threshold cycle (Ct) method. The quantity of specific RNA was calculated using the equation: relative quantity $(RQ) = 2^{-\triangle\triangle CT}$.

**ELISA assay**. To determine the distribution of LIGHT and chemoattractants respond for the lymphocyte infiltration, ELISA assay was performed. For the LIGHT identification (AKR-130, Cell Biolabs), mice were sacrificed to harvest major organs and tumors 24 or 72 h after the administration. For CCL21 and CXCL13 determination (DY457 and MCX130, R&D System), tumors were extracted 24 h after endpoint of the treatments. All samples were processed and analyzed according to manufacturer's instructions. The results were obtained using a microplate spectrophotometer.

**In vivo real-time imaging**. Pancreatic tumor-bearing nude mice were used to evaluate the homing ability, biodistribution and tumor accumulation of nanoparticles in vivo. In brief, KPC1199 $(1 \times 10^6)$ were orthotopically inoculated to the tail of the pancreas of 6-weeks male nude C57BL/6 mice to establish the PDAC model. Different nanoparticles were dosing when the tumor volume reached to 100 mm$^3$. For the in vivo homing ability, DiR-labeled Nano-sapper (DiR = 1 mg/kg) was i.v. injected and imaged 12 h later. For the biodistribution and tumor accumulation, mice were randomly divided into four groups and subjected to different treatments. DiR-labeled CaP (DiR = 1 mg/kg) was administered at day 11, 14, and 18. The locations of tumors were initially confirmed by palpation of the prominent part. Images were acquired and semi-quantitatively analyzed by IVIS imaging system (Caliper Perkin Elmer, USA).

**Establishment of orthotopic PDAC models**. KPC1199 $(1 \times 10^6)$ or Panc02 $(1 \times 10^6)$ cells were orthotopically inoculated into the tail of the pancreas of 6-weeks male C57BL/6 mice to establish the model. When the orthotopic tumors grew to around 100 mm$^3$, the C57BL/6 mice were divided randomly into six groups $(n = 12)$. Group 1: saline (i.v. injection); Group 2: Gem (i.v. injection, 15 mg/kg); Group 3: α-PD-1 (i.p. injection, 200 μg per mouse); Group 4: Nano-sapper (i.v. injection, MP = 13.9 mg/kg; 25 μg plasmid per mouse; 200 μg per mouse of IgG were given synchronized with group 3); Group 5: Nano-sapper + α-PD-1 (i.v. injection, MP = 13.9 mg/kg; 25 μg plasmid per mouse; 200 μg per mouse of α-PD-1 were given synchronized with group 3).

**Blood chemistry and safety analysis**. To assess the adverse effects of the nanoparticles during the treatment, three randomly selected mice in each group were subjected to a toxicity assay a week after the final treatments. Both whole blood and serum were collected. Whole blood cellular components were counted and compared. Serum AST, ALT, and BUN in the serum were assayed. Organs including the heart, liver, spleen, lungs, and kidneys were collected and fixed for H&E staining.

**Statistical analysis**. All data were presented as mean ± standard deviation (SD). Unpaired student's *t*-test was used for between two-group comparison and one-way ANOVA with Bonferroni tests for multiple-group analysis. Differences in survival were determined for each group by the Kaplan–Meier method and *P* value was calculated by the log-rank test. Differences were considered statistically significant if $p < 0.05$.

**Reporting summary**. Further information on research design is available in the Nature Research Reporting Summary linked to this article.

## Data availability

The source data underlying Figs. 2h, 3e, 5c, 9b–c, Supplementary Figs. 10d–f and 14b are provided as a Source Data file. All the data supporting the findings of this study are available within the article and its supplementary information files.

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

## Acknowledgements

This work was supported by National Natural Science Foundation of China (No. 81872498, 81690263, 81673019, 81573382, 81722043, 81973272), National Science and Technology Major Project (2018ZX09734005, 2017ZX09304016), grant from Shanghai Science and Technology Committee (19410710100), National Youth Talent Support Program, and "Shu Guang" project supported by Shanghai Municipal Education Commission and Shanghai Education Development Foundation (15SG14).

## Author contributions

J.C., X.-L.G., and Y.H. designed the research; Y.H. carried out most experiments; Y.C. provided help for most experiments; Y.H., Y.C., and S.Z. analyzed the data; L.C., J.W., Y.P., M.X., J.F., T.J., K.L., and S.L. assisted with the establishment of PDAC-bearing mice model; Q.S. provided help for scientific drawings; G.J., X.G., and Q.Z. assisted with the tumor frozen microtome. Y.H. and X.-L.G. wrote the paper. All authors discussed the results and commented on the paper.

## Competing interests

The authors declare no competing interests.
