## [Peer Review File · Nature Communications]

Reviewers' comments:

Reviewer #1 (Remarks to the Author):

The manuscript, “Dual-Mechanism Based CTLs Infiltration Enhancement Initiated by Nano-sapper Potentiates Immunotherapy against Immune-Excluded Tumors” puts forth a promising nanoparticle based approach to improve the function of the immune compartment in pancreatic cancer. A major challenge in treating this disease is the protective effect of the dense stroma and defective vasculature that surround the tumor. This barrier not only limits delivery of small molecule drugs but also prevents cytotoxic lymphocytes from infiltrating the tumor – thereby also limiting the efficacy of immunotherapeutic approaches. In this paper, the authors developed a nanoparticle (Nano-sapper) that targets pancreatic tumors (via interaction of FHK with tenascin-C), disrupts the fibrotic stroma (via CAF remodeling induced by α -mangostin), and potentiates intratumoral inflammation (via plasmid encoding LIGHT cytokine). Nano-sapper was shown to improve both plasmid delivery and T cell infiltration to pancreatic tumor. Moreover, in orthotopic pancreatic tumor mouse models, treatment with Nano-sapper prolonged survival and inhibited tumor growth. These results were further enhanced when Nano-sapper was combined with ICB (anti-PD1 antibody), suggesting the nanoparticles indeed improve access and activation of T cells in the tumor microenvironment. Overall this work was well investigated and has potential for combination and improvement of a variety of existing therapies.

Some points to address include:

- Scheme 1: Language should be revised from lines 122-128 such that the schematic is a proposed mechanism and not a definitive illustration of what is taking place. Also, it should not be referenced as evidence/data for the observations observed.
- Figure 1F: Visually the number of transfected cells (1E) looks quite similar so claiming higher transfection efficiency by Nano-sapper vs. lipofectamine does not seem appropriate. That being said – the corresponding flow cytometry data (1F) suggests there are perhaps more GFP copies per cell when transfected with Nano-sapper (as indicated by MFI). To confirm this – qPCR can be performed. Otherwise, 1F may be better suited as supplemental data. It should also be clarified what the control is in this experiment and explicitly stated how the data was collected/quantified
- Figure 3B (right side): It is unclear how the mean vessel length data was generated.
- Figure 5F: As a reference for comparison, similar images should be taken in untreated mice to demonstrate the changes induced by Nano-sapper treatment.

Reviewer #2 (Remarks to the Author):

Huang et al: Dual-Mechanism Based CTLs Infiltration Enhancement Initiated by Nano-sapper Potentiates Immunotherapy against Immune-Excluded Tumors

Summary

The authors generated Nanoparticles for immunosensitization of immune-excluded (cold) PDAC that were induced by orthotopic transplantation of isogenic cell lines into immunocompetent host mice. The particles called Nano-sapper represented lipid vesicles, coated with a Tenascin C peptide and a core of Calcium phosphate, Mangosin phosphate and an expression vector for LIGHT. The Tenascin C peptide was used for homing of the Nano-sapper particles into the PDAC, which have a Tenascin C-rich stroma. The Mangosin phosphate normalized tumor blood vessels and reduced the amount of stroma in the tumors thereby removing a physical barrier for immune infiltration. Expression of the cytokine LIGHT induced CTL infiltration and other immunological consequences that converted cold into hot PDAC, which sensitized them to anti PD-1 immunotherapy and interfered with liver metastasis.

General comment

The immunosensitization strategy described in the manuscript is quite imaginative and interesting. Homing of the Nano-sapper particles in the tumors was greatly improved by the Tenascin C peptide and overall toxicity was low. The immunologic consequences of Nano-sapper treatment are impressive and might have therapeutic potential. The manuscript is well designed and a revised version should be considered for publication in Nature Communications.

Major comments

1. A drawback of the study is the use of transplanted tumor cells. Many studies have shown misleading beneficial effects of drugs in pre-clinical tumor transplantation models that did not hold true in clinical studies. There are several Kras-based models for PDAC available that could be used for Nano-sapper treatment. Have the authors considered to test Nano-sapper/anti PD-1 therapy in an autochthonous PDAC model?

2. The authors conclude that sensitizing immune effects are mainly due to LIGHT expression but a suitable control is missing. An alternative explanation would be DNA-mediated activation of cGAS-

Sting. Use of a LIGHT expression vector with a nonsense mutation in the LIGHT coding sequence would discriminate between these possibilities.

Minor comments

1. The manuscript needs extensive proofreading.
2. Individual components of the Nano-sapper such as Light should be described in the introduction but the part about the Nano-sapper design (including Scheme 1) should be moved to the results as Figure 1 - Expected effects of Nano-sapper.
3. There are relatively long introductions in the results section with redundant information (for example in the first paragraph of "Nano-sapper attenuated the stromal barrier in PDAC" where effects of the ECM density on CTL infiltration is explained again. A comprehensive description in the introduction and deletion of redundant information would shorten the manuscript and make it easier to read.
4. Axes in Figure 2a, 4a, 6a, 7a should be labelled.
5. The Western blot in Supplementary Figure 10f is not convincing.
6. The white tracks in Figure 3a should be removed and replaced by arrows.
7. The image in Figure 7e might be a mirror image.
8. The weight of mice should be included in Figure 8.

Response to the referees' comments

Reviewer #1

The manuscript, “Dual-Mechanism Based CTLs Infiltration Enhancement Initiated by Nano-sapper Potentiates Immunotherapy against Immune-Excluded Tumors” puts forth a promising nanoparticle-based approach to improve the function of the immune compartment in pancreatic cancer. A major challenge in treating this disease is the protective effect of the dense stroma and defective vasculature that surround the tumor. This barrier not only limits delivery of small molecule drugs but also prevents cytotoxic lymphocytes from infiltrating the tumor – thereby also limiting the efficacy of immunotherapeutic approaches. In this paper, the authors developed a nanoparticle (Nano-sapper) that targets pancreatic tumors (via interaction of FHK with tenascin-C), disrupts the fibrotic stroma (via CAF remodeling induced by α -mangostin), and potentiates intratumoral inflammation (via plasmid encoding LIGHT cytokine). Nano-sapper was shown to improve both plasmid delivery and T cell infiltration to pancreatic tumor. Moreover, in orthotopic pancreatic tumor mouse models, treatment with Nano-sapper prolonged survival and inhibited tumor growth. These results were further enhanced when Nano-sapper was combined with ICB (anti-PD1 antibody), suggesting the nanoparticles indeed improve access and activation of T cells in the tumor microenvironment. Overall this work was well investigated and has potential for combination and improvement of a variety of existing therapies.

Some points to address include:

1. Scheme 1: Language should be revised from lines 122-128 such that the schematic is a proposed mechanism and not a definitive illustration of what is taking place. Also, it should not be referenced as evidence/data for the observations observed.

Response: Thank you very much for the suggestions. Accordingly, we have revised the description to make sure it focuses on the elaboration of the proposed mechanism. “Considering the similarity in functions, we named FHK-pLIGHT@CaMP as “Nano-sapper”, because sappers have similar responsibilities in breaking through defensive barriers, repairing roads and bridges, building military bases, and finally paving the way for subsequent troops during the war. As described in Scheme 1, the proposed dual-mechanism based infiltration enhancement strategy, involving obstacles reduction and recruiting signals stimulation, would provide a promising approach in sensitizing IET to ICB therapy.” Please find the revised content in Line 117-122 (Page 5).

2. Figure 1F: Visually the number of transfected cells (1E) looks quite similar so claiming higher transfection efficiency by Nano-sapper vs. lipofectamine does not seem appropriate. That being said - the corresponding flow cytometry data (1F) suggests there are perhaps more GFP copies per cell when transfected with Nano-sapper (as indicated by MFI). To confirm this – qPCR can be performed. Otherwise, 1F may be better suited as supplemental data. It should also be clarified what the control is in this experiment and explicitly stated how the data was collected/quantified.

Response: Thank you very much for the concern. As suggested, we have also evaluated the expression of EGFP at mRNA level through RT-qPCR analysis. As shown in the revised Fig. 1f, EGFP-Nano-sapper exhibited similar transfection efficiency with the commercial reagent at the mRNA level despite that flow cytometry analysis (Supplementary Fig. 13b) showed that EGFP-Nano-sapper exhibited slightly higher transfection efficiency at the protein level. One possible explanation is that mRNA do not correlate with protein in a strictly linear manner because the

asynchronism of transcription and translation.¹ As suggested, the flow cytometry results have been moved to Supplementary Fig. 13b to avoid the unintentional exaggeration. Related revision please see Line 169-172 (Page 8).

Additionally, methods concerning transfection assay and data processing has been supplemented in Line 600-612 (Page 29). Specifically, the negative control (NC) was the nanoparticle loaded with non-EGFP coding vector plasmid (FHK-pVector@CaMP), and the positive control was the commercial reagent, lipofectamine LTX with PLUS™ Reagent. Related description please see Line 188-190 (Page 10) and Line 603-606 (Page 29) in the revised manuscript.

Fig. 1f *In vitro* transfection efficiency of EGFP-Nano-sapper on activated NIH3T3 and KPC1199. Transfection efficiency was determined by RT-qPCR analysis of the mRNA level of EGFP 24 h after transfection. The non-EGFP coding vector plasmid loaded nanoparticle (FHK-pVector@CaMP) was used as the negative control (NC) and the commercial reagent is lipofectamine LTX with PLUS™ Reagent (n =3). Data are presented as mean ± s.d., * $p < 0.05$, ** $p < 0.01$, and analyzed by one-way ANOVA with Bonferroni multiple comparisons post-test.

Supplementary Fig. 13b The transfection efficiency determined by flow cytometry. Negative control (NC): the non-EGFP coding vector plasmid loaded nanoparticle (FHK-pVector@CaMP). Data are presented as mean \pm s.d. (n=3). *** p <0.001, analyzed by one-way ANOVA with Bonferroni multiple comparisons post-test.

3. Figure 3B (right side): It is unclear how the mean vessel length data was generated.

Response: Thank you for the comment. The measurement was performed according to two published works (*Cell Rep.* 2015; 13(12):2687-98 and *Nat Immunol.* 2017; 18(11):1207-17), and the detailed description about the method has now been incorporated in the revised manuscript, please see Line 566 to 568 (Page 28). “The mean vessel length was quantified as previously described by using ImageJ to measure the length of continuous CD31 positive signal in the enlarged field of view, and then divided it by the number of vessels.^{2, 3}”

4. **Figure 5F:** As a reference for comparison, similar images should be taken in untreated mice to demonstrate the changes induced by Nano-sapper treatment.

Response: Thank you for the suggestion. Accordingly, images taken in the untreated mice have now been incorporated in the revised Fig. 5F (Page 17). It was found that unlike the phenomenon observed in the Nano-sapper-treated tumor (right), only few infiltrating lymphocytes and no typical spatial feature of TLSs were observed in the untreated control (left).

Fig. 5f Characterization of TLSs in the tumor from the untreated or Nano-sapper treated mice. White bars represent 100 μ m.

Reviewer #2 (Remarks to the Author):

Huang et al: Dual-Mechanism Based CTLs Infiltration Enhancement Initiated by Nano-sapper Potentiates Immunotherapy against Immune-Excluded Tumors

Summary

The authors generated Nanoparticles for immune-sensitization of immune-excluded (cold) PDAC that were induced by orthotopic transplantation of isogenic cell lines into immunocompetent host mice. The particles called Nano-sapper represented lipid vesicles, coated with a Tenascin C peptide and a core of Calcium phosphate, Mangostin phosphate and an expression vector for LIGHT. The Tenascin C peptide was used for homing of the Nano-sapper particles into the PDAC, which have a Tenascin C-rich stroma. The Mangosin phosphate normalized tumor blood vessels and reduced the amount of stroma in the tumors thereby removing a physical barrier for immune infiltration. Expression of the cytokine LIGHT induced CTL infiltration and other immunological consequences that converted cold into hot PDAC, which sensitized them to anti PD-1 immunotherapy and interfered with liver metastasis.

General comment

The immune-sensitization strategy described in the manuscript is quite imaginative and interesting. Homing of the Nano-sapper particles in the tumors was greatly improved by the Tenascin C peptide and overall toxicity was low. The immunologic consequences of Nano-sapper treatment are impressive and might have therapeutic potential. The manuscript is well-designed and a revised version should be considered for publication in Nature Communications.

Major comments

1. A drawback of the study is the use of transplanted tumor cells. Many studies have shown misleading beneficial effects of drugs in pre-clinical tumor transplantation models that did not hold true in clinical studies. There are several Kras-based models for PDAC available that could be used for Nano-sapper treatment. Have the authors considered to test Nano-sapper/anti PD-1 therapy in an autochthonous PDAC model?

Response: Thank you very much for the comments. We agree with the reviewer that an appropriate model is essential to the translation from the lab to the clinic. In this research, we employed the widely used KPC1199 orthotopic model to evaluate the therapeutic potential of the Nano-sapper treatment because it can recapitulate the clinical features of human conditions.^{4, 5, 6} (1) Its similar histopathological features with human PDAC. PDAC is characterized by highly desmoplastic stroma and the duct formation. As shown in Supplementary Fig. 20a, we found the duct-like structures (indicated by yellow dash) which were surrounded by dense stroma, the compressed vessels, and the abnormally activated fibroblasts (characterized by α -SMA, fibronectin, FAP and collagen). (2) Its shared genomic features with human PDAC. Primary tumor cell line KPC1199 is derived from KPC pancreatic ductal adenocarcinoma mouse (LSL-*Kras*^{G12D/+}; LSL-*Trp53*^{R172H/+}; *Pdx-1-Cre*, on C57BL/6 background) and possess *Kras* and *p53* mutation. Mutations in both *Kras* and *p53* genes are found in around 90% and 75% of all human PDAC respectively.⁷ Thus, the *Kras* and *p53* double-mutation in KPC1199 is able to reflect the most prevalent genotype of PDAC. (3) Its similar inflammatory features with human PDAC development. Pancreatitis is indispensable to PDAC development,⁸ and interleukin 6 (IL-6) is responsible for the progression from chronic pancreatitis to pancreatic cancer.⁹ Thus, we characterized the inflammatory feature of the KPC1199 orthotopic model through determining the level of IL-6. As shown in Supplementary Fig. 20b, IL-6 is

extensively expressed in the KPC1199 orthotopic model, which resembles the phenomenon in clinic. Moreover, both the α -PD-1 and gemcitabine treatments did not influence the IL-6 level which was significantly decreased by Nano-sapper and Nano-sapper combined with α -PD-1. (4) Its similar therapeutic features with human PDAC. There is more than 80% of patients with PDAC have frankly metastasis accompany with the primary tumor and unusually resistant to all forms of cytotoxic chemotherapies.¹⁰ In this work, we observed the liver metastasis and the gemcitabine treatments were greatly compromised in KPC1199 orthotopic model (Fig. 6). From the above points of view, we believe KPC1199 might serve as a reasonable model here. In addition, considering the genetic diversity of PDAC in clinic, we also evaluated the effect of Nano-sapper in another PDAC transplantation model with different mutations. Panc02 cell line is an established Grade III adenocarcinoma model developed by chemical induction with 3-MCA (3-methylcholanthrene), which has both *Kras* and *Smad4* (occurred in around 50% of pancreatic cancers) mutations.^{11, 12, 13} As shown in Fig 7, Nano-sapper combined with α -PD-1 also significantly prolonged the survival of Panc02-bearing mice, implying its potential in treating PDAC with heterogeneous genotypes.

We agree with the reviewer that cautions remain necessary in translating the information generated by transplantation models to human pancreatic cancer. Although the models we used cover the key pathological and genetic features of the human PDAC, we agree that autochthonous PDAC model, whose large-scale preclinical application usually suffer from long latency periods (usually > 4 months to generate tumors) and big variety,^{14, 15} can be applied as an auxiliary tool to evaluate the effect of our strategy in future work. Relevant discussion has been incorporated in the revised manuscript, please see Line 428-443 (Page 24-25).

Supplementary Fig. 20 Histopathological features of KPC1199 model and IL-6 in TME was downregulated by Nano-sapper. (a) Histopathological characterization of KPC1199 model. (b) Nano-sapper reduced the expression of IL-6. White bars represent 100 μ m. Data are presented as mean \pm s.d.. * p <0.05, *** p <0.001, analyzed by one-way ANOVA with Bonferroni multiple comparisons post-test.

2. The authors conclude that sensitizing immune effects are mainly due to LIGHT expression, but a suitable control is missing. An alternative explanation would be DNA-mediated

activation of cGAS-Sting. Use of a LIGHT expression vector with a nonsense mutation in the LIGHT coding sequence would discriminate between these possibilities.

Response: Thanks for the valuable comments. Indeed, the incorporated plasmids might also sensitize TME through the cGAS-STING signaling pathway. To test this possibility, we additionally prepared nanoparticle loaded with scrambled LIGHT sequences-encoded vector plasmid (FHK-pVector@CaMP), and treated the KPC1199-bearing mice with this formulation (Supplementary Fig. 19a). It was found that neither signals of CD8 (IHC) nor amount of CD45⁺CD3⁺CD8⁺ T cells was enhanced following the FHK-pVector@CaMP treatment compared with that after the FHK-CaMP treatment (Supplementary Fig. 19b and c), indicating that the nonsense plasmid did not induce observable changes in T cell infiltration. Two possible reasons might account for these phenomena. First, the compaction by protamine reduced the exposure of plasmid to cGAS. Protamine sheltered the pentose-phosphate backbone of plasmid through the interaction between positively charged amino acid residues of protamine and negatively charged phosphate groups of plasmids.¹⁶ Since the backbone of double-stranded DNA happens to be the binding site of cGAS,¹⁷ the activation of cGAS-STING could be relieved by protamine. Second, STING activation could possibly be counteracted by α -M. Actually, α -M have been reported to decrease the activation of several signaling pathways including IL-1 and IL-6, which are inflammatory cytokines driven by cGAS-STING activation.¹⁸ In general, the immune-sensitizing activity of nonsense plasmid is not obvious in our current study. Therefore, we concluded that the sensitizing immune effects of Nano-sapper are mainly due to LIGHT expression. Relevant discussion please see Line 415-427 (Page 24) in the revised manuscript.

Supplementary Fig. 19 The loaded non-sense plasmid did not induce enhanced infiltration of CD8⁺ T cell. (a) Six-week-old male C57BL/6 mice were orthotopically inoculated with KPC1199 cells (1×10^6) on day 0. FHK-CaMP (MP = 13.9 mg/kg), FHK-pVector@CaMP (MP = 13.9 mg/kg, 25 μg plasmid per mouse) were *i.v.* administrated every other day (n = 3). At day 16, tumors were extracted and subjected to IHC and flow cytometry assays. Images are collected from randomly selected ten fields of visions. FHK-pVector@CaMP: nanoparticle loaded with scrambled LIGHT sequences-encoded vector plasmid. Data are presented as mean \pm s.d., and analyzed by one-way ANOVA with Bonferroni multiple comparisons post-test. White bars represent 100 μm.

Minor comments

1. The manuscript needs extensive proofreading.

Response: Thanks for your suggestion. We have made extensive proofreading throughout the manuscript and made corrections accordingly. All the corrections were highlighted in yellow in the revised manuscript.

2. Individual components of the Nano-sapper such as Light should be described in the introduction, but the part about the Nano-sapper design (including Scheme 1) should be moved to the results as Figure 1 - Expected effects of Nano-sapper.

Response: Thank you for the comments. Accordingly, we have described the components of Nano-sapper including α -M phosphate, LIGHT coding plasmid, FHK-peptide in the introduction from Line 84-105 (Page 4-5). The part about the Nano-sapper design was moved the results of “**The preparation and characterization of Nano-sapper**” in Line 149-155 (Page 7) and the caption of scheme 1 was revised as “**Scheme 1** The expected effects of MP and pLIGHT co-loaded FHK-decorated calcium phosphate liposome (FHK-pLIGHT@CaMP, Nano-sapper) combined with α -PD-1 (anti-programmed cell death protein 1)” which could be found in Line 125-127 (Page 6).

3. There are relatively long introductions in the results section with redundant information (for example in the first paragraph of “Nano-sapper attenuated the stromal barrier in PDAC” where effects of the ECM density on CTL infiltration is explained again. A comprehensive description in the introduction and deletion of redundant information would shorten the manuscript and make it easier to read.

Response: Thank you for the suggestion. We have deleted the redundant information in results part of the revised manuscript. Revised content could be found in Line 149 (Page 7), Line 203 (Page 10) and Line 232-233 (Page 12).

4. Axes in Figure 2a, 4a, 6a, 7a should be labelled.

Response: Thank you for the suggestion. We have labelled the axes in Fig. 2a, 4a, 6a, 7a and Supplementary Fig. 15a in the revised version.

Fig. 2a Six-week-old male C57BL/6 mice were orthotopically inoculated with KPC1199 cells (1×10^6) on day 0. Saline, FHK-CaMP (MP = 13.9 mg/kg), FHK-pLIGHT@CaP (25 μ g plasmid per mouse) and Nano-sapper (MP = 13.9 mg/kg, 25 μ g plasmid per mouse) were *i.v.* administrated on day 11, 13 and 15. On day 16, tumors were harvested for analysis.

Fig. 4a Six-week-old male C57BL/6 mice were orthotopically inoculated with KPC1199 cells (1×10^6) on day 0. Saline, FHK-CaMP (MP = 13.9 mg/kg), FHK-pLIGHT@CaP (25 μ g plasmid per mouse) and Nano-sapper (MP = 13.9 mg/kg, 25 μ g plasmid per mouse) were *i.v.* administrated on day 11, 13, 15 and 17, and DiR-labeled CaP was *i.v.* administrated on day 11, 14 and 18 (DiR = 1 mg/kg), and subjected to IVISTM for *in vivo* imaging. On day 18, tumors were harvested for *ex vivo* detection.

Fig. 6a Six-week-old male C57BL/6 mice were orthotopically inoculated with KPC1199 cells (1×10^6) on day 0. Saline, Gem (15 mg/kg) and Nano-sapper (MP = 13.9 mg/kg, 25 μ g plasmid per mouse) were *i.v.* administrated every other day and α -PD-1 or IgG isotype (200 μ g per mouse) were *i.p.* administrated every two days (n = 12).

Fig. 7a Six-week-old male C57BL/6 mice were orthotopically inoculated with Panc02 (1×10^6) on day 0. Saline, Gem (15 mg/kg) and Nano-sapper (MP = 13.9 mg/kg, 25 μ g plasmid per mouse) were *i.v.* administrated every other day and α -PD-1 or IgG isotype (200 μ g per mouse) were *i.p.* administrated every two days (n = 12).

Supplementary Fig. 15a Six-week-old male C57BL/6 mice were orthotopically inoculated with KPC1199 cells (1×10^6) on day 0. Saline and FHK-CaMP with different MP dosage (6.8, 10.2 or 13.9 mg/kg) were *i.v.* administrated on day 11, 13 and 15. On day 16, tumors were harvested and subjected to IHC analysis and Masson staining.

5. The Western blot in Supplementary Figure 10f is not convincing.

Response: We have semi-quantified the images through ImageJ and supplemented the results in the revised Supplementary Fig. 10g. Briefly, MP treatments reversed the increments of pSmad2/3 expression that promoted by TGF- β , and MP of 27.8 μ M resulted the maximum inhibition.

Supplementary Fig. 10g Phosphorylation of Smad2 and Smad3 in activated NIH3T3 were inhibited after incubation with MP for 24 h. Data are presented as mean \pm s.d.. (n=3), *** p <0.001, analyzed by one-way ANOVA with Bonferroni multiple comparisons post-test.

6. The white tracks in Figure 3a should be removed and replaced by arrows.

Response: Thank you for pointing this out, we have replaced white tracks with arrows in Fig 3a of the revised manuscript.

Fig. 3a Tumor slices were stained with anti-CD31 for blood vessel characterization. White bars represent 100 μ m.

7. The image in Figure 7e might be a mirror image.

Response: Thank you very much for pointing out our mistake. We have corrected it. Revision please see Fig. 7d in the revised manuscript.

Fig. 7d Tumor inhibition of different treatments in Panc02 bearing mice (n = 5).

8. The weight of mice should be included in Figure 8.

Response: We have included the weight of mice in the revised Fig. 8b.

Fig. 8b The weight of mice monitored during the therapy. Data are presented as mean \pm s.d. (n = 5).

Reference

1. Gedeon T, Bokes P. Delayed protein synthesis reduces the correlation between mRNA and protein fluctuations. *Biophys J* **103**, 377-385 (2012).
2. Johansson-Percival A, *et al.* Intratumoral light restores pericyte contractile properties and vessel integrity. *Cell Rep* **13**, 2687-2698 (2015).

3. Johansson-Percival A, *et al.* De novo induction of intratumoral lymphoid structures and vessel normalization enhances immunotherapy in resistant tumors. *Nat Immunol* **18**, 1207-1217 (2017).
4. He W, *et al.* Il22ra1/stat3 signaling promotes stemness and tumorigenicity in pancreatic cancer. *Cancer Res* **78**, 3293-3305 (2018).
5. Niu N, *et al.* Loss of setd2 promotes kras-induced acinar-to-ductal metaplasia and epithelia–mesenchymal transition during pancreatic carcinogenesis. *Gut*, gutjnl-2019-318362 (2019).
6. Jing W, *et al.* Sting agonist inflames the pancreatic cancer immune microenvironment and reduces tumor burden in mouse models. *J Immunother Cancer* **7**, 115 (2019).
7. Hingorani SR, *et al.* Trp53r172h and krasg12d cooperate to promote chromosomal instability and widely metastatic pancreatic ductal adenocarcinoma in mice. *Cancer Cell* **7**, 469-483 (2005).
8. Shi J, Xue J. Inflammation and development of pancreatic ductal adenocarcinoma. *Chin Clin Oncol* **8**, 19 (2019).
9. Zhang Y, *et al.* Interleukin-6 is required for pancreatic cancer progression by promoting mapk signaling activation and oxidative stress resistance. *Cancer Res* **73**, 6359-6374 (2013).
10. Yang L, *et al.* Overexpression of fzd1 and caix are associated with invasion, metastasis, and poor-prognosis of the pancreatic ductal adenocarcinoma. *Pathol Oncol Res* **24**, 899-906 (2018).
11. Corbett TH, *et al.* Induction and chemotherapeutic response of two transplantable ductal adenocarcinomas of the pancreas in c57bl/6 mice. *Cancer Res* **44**, 717-726 (1984).
12. Wang Y, *et al.* Genomic sequencing of key genes in mouse pancreatic cancer cells. *Curr Mol Med* **12**, 331-341 (2012).
13. Cicenias J, Kvederaviciute K, Meskinyte I, Meskinyte-Kausiliene E, Skeberdyte A, Cicenias J. Kras, tp53, cdkn2a, smad4, brca1, and brca2 mutations in pancreatic cancer. *Cancers (Basel)* **9**, (2017).
14. Mazur PK, Siveke JT. Genetically engineered mouse models of pancreatic cancer: Unravelling tumour biology and progressing translational oncology. *Gut* **61**, 1488-1500 (2012).
15. Herreros-Villanueva M, Hijona E, Cosme A, Bujanda L. Mouse models of pancreatic cancer. *World J Gastroenterol* **18**, 1286-1294 (2012).
16. Biegeleisen K. The probable structure of the protamine–DNA complex. *J Theor Biol* **241**, 533-540 (2006).
17. Jiang H, *et al.* Targeting focal adhesion kinase renders pancreatic cancers responsive to checkpoint immunotherapy. *Nat Med* **22**, 851-860 (2016).
18. Gutierrez-Orozco F, Failla ML. Biological activities and bioavailability of mangosteen xanthones: A critical review of the current evidence. *Nutrients* **5**, 3163-3183 (2013).

REVIEWERS' COMMENTS:

REVIEWER #1 COMMENTS

The concerns that were previously raised were all well addressed by the authors, and the follow up work and clarification added acceptable validity to the manuscript.

An additional minor point of concern that should still be fixed is the merged image in Supplemental Figure 14 C. It appears that for the red channel, the overlay of the images presented in the top group (pLIGHT@CaMP) does not resemble the individual panel shown. It looks as though the merged image may have mistakenly incorporated a zoomed out version of the individual image.

REVIEWER #2 COMMENTS

Although the authors have not included data from autochthonous pancreatic tumors, their explanations about the used transplantation system and its obvious close similarity to human PDACs is convincing. Importantly, a potential implication of cGas/Sting was ruled out by additional pre-clinical studies. Western blot data have been improved and weight of mice has been included. Therefore, the reviewer has no further requests.

Response to referees' comments

REVIEWER #1 COMMENTS

The concerns that were previously raised were all well addressed by the authors, and the follow up work and clarification added acceptable validity to the manuscript. An additional minor point of concern that should still be fixed is the merged image in Supplemental Figure 14 C. It appears that for the red channel, the overlay of the images presented in the top group (pLIGHT@CaMP) does not resemble the individual panel shown. It looks as though the merged image may have mistakenly incorporated a zoomed out version of the individual image.

Response: We thank Reviewer #1 for the positive evaluation of our revision and for pointing out our mistakes. We have revised Supplemental Figure 14c in the revised version of our paper.

REVIEWER #2 COMMENTS

Although the authors have not included data from autochthonous pancreatic tumors, their explanations about the used transplantation system and its obvious close similarity to human PDACs is convincing. Importantly, a potential implication of cGas/Sting was ruled out by additional pre-clinical studies. Western blot data have been improved and weight of mice has been included. Therefore, the reviewer has no further requests.

Response: We are grateful to the reviewer for the valuable comments that were provided in all the revision stages.